# TensorVAE: a direct generative model for molecular conformation generation driven by novel feature engineering

## Abstract

Efficient generation of 3D conformations of a molecule from its 2D graph is a key challenge in in-silico drug discovery. Deep learning (DL) based generative modeling has recently become a potent tool to tackling this challenge. However, many existing DL-based methods are either indirect–leveraging inter-atomic distances or direct–but requiring numerous sampling steps to generate conformations. In this work, we propose a simple model abbreviated TensorVAE capable of generating conformations directly from a 2D molecular graph in a single step. The main novelty of the proposed method is *focused on feature engineering*. We develop a novel encoding and feature extraction mechanism relying solely on *standard convolution operation* to generate token-like feature vector for each atom. These feature vectors are then transformed through *standard transformer encoders* under a conditional Variational Autoencoder framework for generating conformations directly. We show through experiments on two benchmark datasets that with intuitive feature engineering, a relatively simple and standard model can provide promising generative capability rivalling recent state-of-the-art models employing more sophisticated and specialized generative architecture.

## 1 Introduction

Recent advance in deep learning has enabled significant progress in computational drug design (Chen et al., 2018). Particularly, capable graph-based generative models have been proposed to generate valid 2D graph representation of novel drug-like molecules (Honda et al., 2019; Mahmood et al., 2021; Yu & Yu, 2022), and there is an increasing interest on extending these methods to generating 3D molecular structures which are essential for structured-based drug discovery (Li et al., 2021; Simm et al., 2021; Gebauer et al., 2022). A stable 3D structure or conformation of a molecule is specified by the 3D Cartesian coordinates of all its atoms. Traditional molecular dynamics or statistical mechanic driven Monte Carlo methods are computationally expensive, making them unviable for generating 3d molecular structures at scale (Hawkins, 2017). In this regard, deep learning(DL)-based generative methods have become an attractive alternative.

DL-based generative methods may be broadly classified into three categories: distance-based, reconstruction-based, and direct methods. The main goal of distance-based methods is learning a probability distribution over the inter-atomic distances. During inference, distance matrices are sampled from the learned distribution and converted to valid 3D conformations through post-processing algorithms. Two representative methods of this category include GraphDG (Simm & Hernández-Lobato, 2019) and CGCF (Xu et al., 2021a). An advantage of modeling distance is its roto-translation invariance property–an important inductive bias for molecular geometry modeling (Köhler et al., 2020). Additional virtual edges and their distances between $2^{nd}$ and $3^{rd}$ neighbors are often introduced to constrain bond angles and dihedral angles crucial to generating a valid conformation. However, Luo et al. (2021) have argued that these additional bonds are still inadequate to capture structural relationship between distant atoms. To alleviate this issue, DGSM (Luo et al., 2021) proposed to add higher-order virtual bonds between atoms in an expanded neighborhood region. Another weakness of the distance-based methods is the error accumulation problem; random noise in the predicted distance can be exaggerated by an Euclidean Distance Geometry algorithm, leading to generation of inaccurate conformations (Xu et al., 2022; 2021b).

To address the above weaknesses, reconstruction-based methods directly model a distribution over 3D coordinates. Their main idea is to reconstruct valid conformations from distorted coordinates. GeoDiff (Xu et al., 2022) and Uni-Mol (Zhou et al., 2022) are pioneering studies in this respect. Though sharing similar idea, they differ in the process of transforming corrupted coordinates to stable conformations. While GeoDiff adapts a reverse diffusion process (Sohl-Dickstein et al., 2015), Uni-Mol treats conformation reconstruction as an optimization problem. Despite their promising performance, both methods require designing of task-specific and complex coordinate transformation methods. This is to ensure the transformation is roto-translation or SE(3)-equivariant. To achieve this, GeoDiff proposed a specialized SE(3)-equivariant Markov transition kernel. On the other hand, Uni-Mol accomplished the same by combining a task-specific adaption of transformer (Vaswani et al., 2017) inspired by the AlphaFold's Evoformer (Jumper et al., 2021) with another specialized equivariant prediction head (Satorras et al., 2021). Furthermore, GeoDiff requires numerous diffusing steps to attain satisfactory generative performance which can be time consuming.

CVGAE (Mansimov et al., 2019) and DMCG (Zhu et al., 2022) have attempted to resolve the generative efficiency issue by developing models that can produce a valid conformation directly from a 2D molecular graph in a single sampling step. Regrettably, the performance of CVGAE is significantly worse than its distance-based counterparts mainly due to the use of inferior graph neural network for information aggregation (Zhu et al., 2022). DMCG aimed to improve the performance of its predecessor by using a more sophisticated graph neural network and a loss function invariant to symmetric permutation of molecular substructures. Although DMCG achieved superior performance, acquiring such loss function requires enumerating all permutations of a molecular graph, which can become computationally expensive for long-sequence molecules.

Regardless of their category, a common recipe of success for these models can be distilled to developing model architecture with ever increasing sophistication and complexity. There is little attention on input feature engineering. In this work, we forgo building specialized model architecture but instead focus on intuitive input feature engineering. We propose to encode a molecular graph using a fully-connected and symmetric tensor. For preliminary information aggregation, we run a rectangle kernel filter through the tensor in a 1D convolution manner. This operation has a profound implication; with a filter size of 3, the information from two immediate neighbors as well as all their connected atoms can be aggregated onto the focal atom in a single operation. It also generates token-like feature vector per atom which can be directly consumed by a standard transformer encoder for further information aggregation.

The generative framework follows the standard conditional variational autoencoder (CVAE) setup. We start with building two input tensors with one encoding only the 2D molecular graph and the other also encoding 3D coordinate and distance. Both tensors go through the same feature engineering step and the generated feature vectors are fed through two separate transformer encoders. The output of these two encoders are then combined in an intuitive way to form the input for another transformer encoder for generating conformation directly. The complete generative model is abbreviated as TensorVAE.

In summary, the proposed method has three main advantages. (1) ***Direct and Efficient***, generating conformation direclty from a 2D molecular graph in a single step. (2) ***Simple***, not requiring task-sepecific design of neural network architecture, relying only on simple convolution and off-the-shelf transformer architecture; (3) ***Easy to implement***, no custom module required as both `PyTorch` and `TensorFlow` offer ready-to-use convolution and transformer implementation. These advantages translate directly to excellent practicality of the TensorVAE method. We demonstrate through extensive experiments on two benchmark datasets that the proposed TensorVAE, despite its simplicity, can perform competitively against **18 recent state-of-the-art methods** for conformation generation and molecular property prediction.

## 2 METHOD

### 2.1 PRELIMINARIES

**Problem Definition.** We formulate molecular conformation generation as a conditional generation task. Given a set of molecular graphs $G$ and their corresponding i.i.d conformations $R$, the goal

is to train a generative model that approximates the Boltzman distribution, and from which a valid conformation conditioned on a molecular graph can be easily sampled in a single step.

**Story Line.** In the ensuing sections, we breakdown the formulation of the proposed method in three novel ideas. We first introduce how the tensor encoding method is derived. Based on the tensor input, we propose a "naive" model that applies a convolutional neural network directly on the tensor to generate distance matrix, from which conformation are obtained through an Euclidean Distance Geometry algorithm. We also show that such model can already provide comparable performance to several advanced distance-based methods. Subsequently, we demonstrate how token-like feature vector can be generated from the input tensor by using 1D convolution operation. Finally, we elaborate on how to combine all the components together under a CVAE framework to arrive at the final generative model.

## 2.2 INPUT TENSOR GRAPH

Graph neural network (GNN) is a popular feature extraction backbone for DL-based molecular conformation generation. The input of GNN in this case is composed of three components, including atom features, edge features and an adjacency matrix. Atom and edge features normally pass through separate embedding steps before being fed to the GNN. Adjacency matrix is then used to determine neighboring atoms for layer-wise information aggregation. Although bond features are aggregated onto atom features and vice versa, these two features are maintained separately throughout the message passing layers (Gilmer et al., 2017; Satorras et al., 2021). Instead of having separate inputs, **our first simple idea** is to combine them into a single input. Specifically, we add an additional dimension to the adjacency matrix, making it a tensor, similar to that used in a computer vision task. The diagonal section of the tensor holds the atom features.

We consider three types of atom features comprising atom type, charge and chirality. Each feature is one-hot encoded and they are stacked together to form a single atom feature vector. There are two variants of the atom feature vector corresponding to two input tensors for the two encoders of the CVAE: an encoder conditioned only on graph (for which the tensor is referred to as the $G$ tensor) and the other conditioned on both graph and coordinates (for which the tensor is referred to as the $GDR$ tensor). For the GDR tenosr, every atom feature vector has three additional channels incorporating the 3D coordinate of the respective atom, and a distance channel filled with zeros.

The off-diagonal section holds the bond features. The considered bond features are bond type, bond stereochemistry type, associated ring size

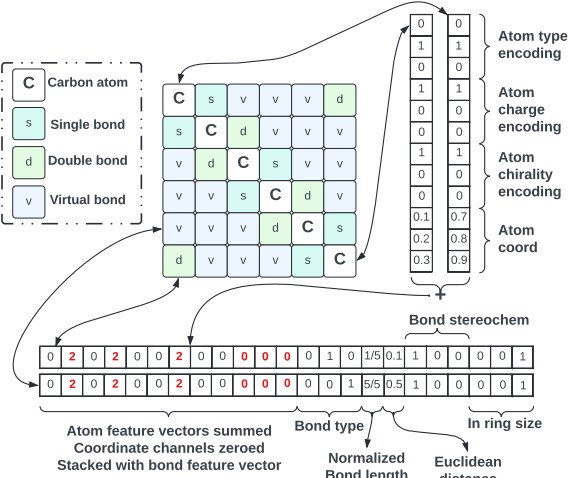

Figure 1: Benzene ring tensor graph example. Note that the values in the feature vector and its dimension are for demonstration purpose only. We explain how they are determined in Sec.3.

and normalized bond length. A virtual bond is also included in the bond type. The normalized bond length is calculated as edge length (1 for direct neighbor, 2 for $2^{nd}$ neighbor, etc.) divided by the longest chain length. It is worth noting that all high-order virtual bonds share the same virtual bond type; they only differ in their normalized bond length. To construct bond feature vector, we first sum the atom feature vectors of the related atoms. This new vector is then stacked with one-hot encoded bond type vector, normalized bond length, and one-hot encoded ring size vector to become the bond feature vector.

There are also two variants of the bond feature vector. For the G tensor, coordinate and distance channles are excluded from all bond feature vectors. For the GDR tensor, to match the size of the

atom feature vector, every bond feature vector has three more coordinate channels filled with 0s, and an additional distance channel holding the Euclidean distance between two connected atoms. This bond feature vector is obtained for all atom pairs, making the proposed tensor fully-connected and symmetric. Despite being referred differently, the structure of the bond feature vector and atom feature vector are the same. For both types in the GDR tensor, there are 5 blocks (3 blocks for G tensor) of channels stacked in the exact same order as following. Therefore, convolution over the input tensor is uniform for both on-diagonal and off-digonal channels.

- atom feature channels (atom type, charge, and chirality);
- atom coordinate channels (coordinate channels are excluded in the G tensor);
- bond type feature channels (bond type and normalized bond length);
- Euclidean distance channel (pairwise distance channel is excluded in G tensor);
- other bond type feature channels (bond stereo-chem type and bond ring size)

An example input tensor graph of the benzene ring is illustrated in Fig.1. Having obtained the tensor representation, a naive way of building a generative model is to apply a convolutional neural network directly on the tensor, and train it to predict a distribution over the inter-atomic distances. We utilize a standard UNet (Ronneberger et al., 2015) structure to map the input tensor to a probability distribution over a distance matrix containing all pair-wise Euclidean distances. Distance matrices are then sampled and converted to valid conformations following the same method presented in GraphDG (Simm & Hernández-Lobato, 2019). We refer to this model as the NaiveUNet. More details of the NaiveUNet can be found in Sec.A.3, and a further explaination of its poor performance can be found in Sec.A.8.

Despite its naive nature, this model achieves a mean coverage (COV) score of $52.14 \pm 1.48\%$ and a mean matching (MAT) score of $1.4322 \pm 0.0247$Å on the GEOM-Drugs dataset, already comparable to several more complex distance-based baselines, as shown in Sec.3.2. However, there are two major issues to this approach. First, with a small kernel size ($3 \times 3$ used in the UNet), it takes many convolution layers to achieve information aggregation between atoms that are far apart; it does not take full advantage of high-order bonds already made available in the input tensor. Secondly, the output size grows quadratically with the number of atoms, as compared to only linear growth in reconstruction-based or direct generation methods. The solution to the first issue is rather simple, obtained by increasing the kernel size to expand its "field of view". On the other hand, solving the second issue requires elevating the naive two-step generative model to a direct one.

### 2.3 Extended kernel and Attention Mechanism

We observe that every row or column of the proposed tensor contains global information of a focal atom and all of its connected atoms (by both chemical and virtual bond). This motivates our **second main idea** which is to extend the length of the kernel to the length of the tensor graph while keeping the width unaltered. This idea has a profound implication; information from the immediate neighbors, all their connected atoms, and all the bond features can be aggregated onto the focal atom in a single convolution operation. In contrast, achieving the same aggregation may require many layers of propagation for the naive model and other GNN-based models. A direct consequence of this modification is that only 1D convolution is permitted. With multiple kernels being applied simultaneously, each stride of these kernels generates a feature vector for a single atom. An illustration of the 1D convolution operation is shown in Fig.2.

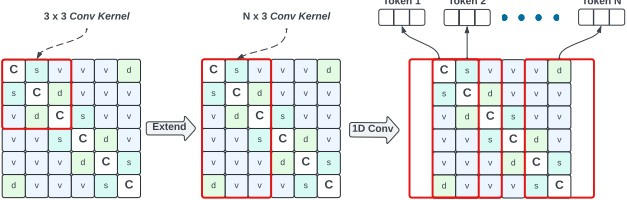

Figure 2: Extending kernel and 1D convolution.

We further observe that the generated feature vectors resemble the token-like feature vectors used in language modeling. This observation combined with the proven success of attention mechanism

in other related work leads to the selection of transformer architecture as the backbone of our generative model. A significant advantage of using transformer's self-attention mechanism is, similar to the extended kernel, it enables a global information aggregation from and for all atoms. It also eliminates the need to maintain separated atom and bond features at each step of feature transformation. We present further insight and a more detailed analysis of the adavantage of this input feature engineering in Sec.A.1

## 2.4 PUTTING EVERYTHING TOGETHER

**Conditional variational autoencoder framework.** We aim at obtaining a generative model $p_\theta(R|G)$ that approximates the Boltzmann distribution through Maximum Likelihood Estimation. Particularly, given a set of molecular graphs $G$ and their respective ground-truth conformations $R$, we wish to maximize the following objective.

$$\log p_\theta(R|G) = \log \int p(z) p_\theta(R|z, G) \, dz \tag{1}$$

A molecular graph can have many random conformations. We assume this randomness is driven by a latent random variable $z \sim p(z)$, where $p(z)$ is a known distribution e.g. a standard normal distribution. As $p_\theta(R|z, G)$ is often modeled by a complex function e.g. a deep neural network, evaluation of the integral in Eq.1 is intractable. Instead, we resort to the same techniques proposed in the original VAE (Kingma & Welling, 2013) to establish a tractable lower bound for Eq.1.

$$\log p_\theta(R|G) \geq \mathbb{E}_{q_w(z|R,G)}[\log p_\theta(R|z, G)] - D_{KL}[q_w(z|R, G)\,||\,p(z)] \tag{2}$$

where $D_{KL}$ is the Kullback-Leibler divergence and $q_w(z|R, G)$ is a variational approximation of the true posterior $p(z|R, G)$. We assume $p(z) = \mathcal{N}(0, \boldsymbol{I})$ and $q_w(z|R, G)$ is a diagonal Gaussian distribution whose means and standard deviations are modeled by a transformer encoder. The input of this transformer encoder is the proposed tensor containing both the coordinate and distance information. We denote this tensor the GDR tensor. On the other hand, $p_\theta(R|z, G)$ is further decomposed into two parts: a decoder $p_{\theta_2}(R|z, \sigma_{\theta_1}(G))$ for predicting conformation directly and another encoder $\sigma_{\theta_1}(G)$ for encoding the 2D molecular graph. The input tensor for $\sigma_{\theta_1}(G)$ is absent of coordinate and distance information, and is therefore denoted the G tensor. Both encoders share the same standard transformer encoder structure. However, there is a minor modification to the transformer structure for the decoder. Specifically, the Query, Key matrices for the first multi-head attention layer are computed based on the output vectors of $\sigma_{\theta_1}(G)$, and the Value matrices come directly from the reparameterization of the output of $q_w(z|R, G)$, as $z = \mu_w + \Sigma_w \epsilon$, where $\mu_w$ and $\Sigma_w$ are the predicted mean and standard deviation respectively. $\epsilon$ is sampled from $\mathcal{N}(0, \boldsymbol{I})$. We present the complete picture of how the two encoders and the decoder are arranged in a CVAE framework in Fig.3a. We also show the illustration of the modified multi-head attention in Fig.3b.

==Intuition behind the modified attention.== There are multiple ways to join together the output of the two encoders to form the input to the final decoder. Popular methods include stacking or addition. We tried both these methods with unsatisfactory performance. We notice that, due to direct stacking or addition of the sampled output of $q_w$ onto the output of $\sigma_{\theta_1}$, attention weights computed in the first layer of the decoder are easily overwhelmed by random noise of the sampled values, and become almost indiscernible[1]. This leads to ineffective information aggregation which is then further cascaded through the remaining attention layers. Intuitively, in the first attention layer, the attention weights dictating how much influence an atom exerts on the other should predominantly be determined by the graph structure, and remain stable for the same molecule. Further, attention weights are computed by Query and Key matrices. Therefore, these two matrices should stay stable for the same graph. This motivates **our third and final main idea**; that is, we compute Query and Key matrices only from the output $\{h_1^L, ..., h_N^L\}$ of $\sigma_{\theta_1}$, and attribute the variation in conformation to the Value matrices which are directly sampled from $\{z_1, ..., z_N\} \sim q_w$. The resultant information aggregation is much more meaningful and each output vector corresponding to an individual atom carries distinct features, facilitating information aggregation of the ensuing attention layers.

---

[1]Imagine a mixture model with randomly varying mixture weights.

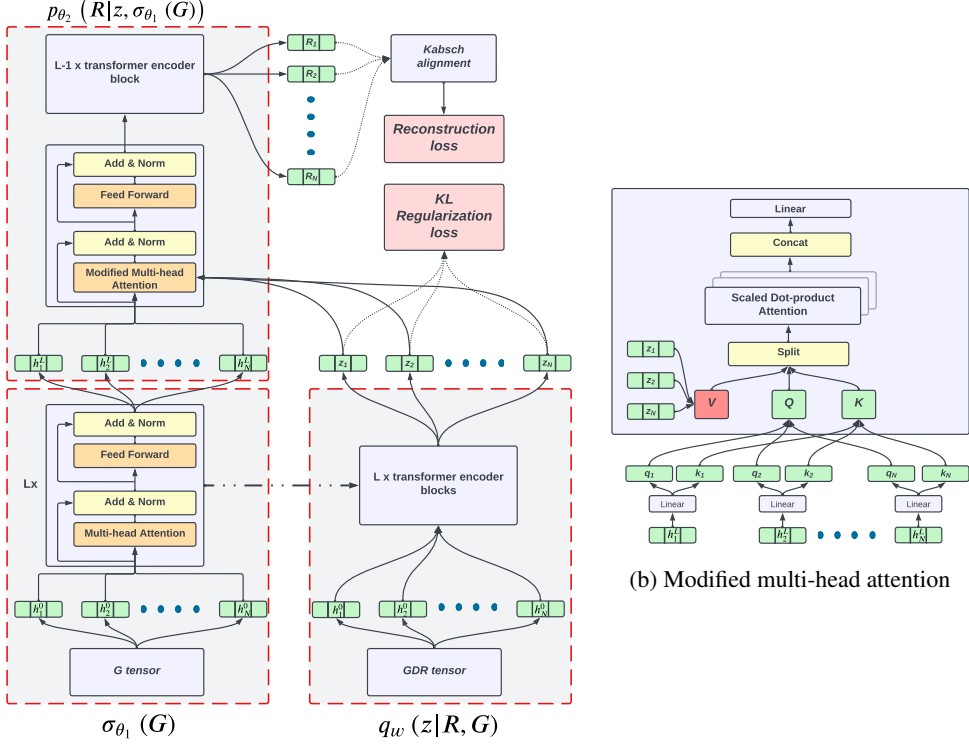

(a) Variational AutoEncoder framework

(b) Modified multi-head attention

Figure 3: TensorVAE model

**Roto-translation invariant loss.** Following ConfVAE (Xu et al., 2021b), we formulate the reconstruction loss as.

$$L\left(R\right) = -\log p_\theta\left(R|z, G\right) = -\sum_{i=1}^{N}\sum_{j=1}^{3}\left(R_{ij} - A\left(\hat{R}, R\right)_{ij}\right)^2 \tag{3}$$

where $A\left(\cdot\right)$ is a function aligning the predicted conformation $\hat{R}$ onto the reference conformation $R$. We choose Kabsch algorithm (contributors, 2022) as the alignment method which translates and rotates the predicted conformation onto its corresponding ground-truth before loss computation. This makes the reconstruction loss roto-translation invariant. Finally, the KL-loss component $D_{KL}\left[q_w\left(z|R, G\right)||p\left(z\right)\right]$ does not involve any coordinate. Therefore, the objective function defined in Eq.2 is roto-translation invariant.

**Direct conformation generation at inference time.** To generate a single conformation, we first construct the G tensor of a molecular graph and obtain a single latent sample $\{z_1, ...z_N\}$ from a standard diagonal Gaussian distribution. The G tensor is passed through $\sigma_{\theta_1}$ encoder to produce $\left\{h_1^L, ..., h_N^L\right\}$ which is then combined with the latent sample via the modified multi-head attention mechanism. The output of this modified attention layer further goes through $L-1$ standard attention layers to be transformed to the final conformation. The entire generation process depends only on a 2D molecular graph, and requires a single sampling step and a single pass of the TensorVAE model.

## 3 EXPERIMENT

In this section, we first elaborate on the implementation details of the TensorVAE model including determining the size of the input tensors, network architecture and how the entire framework is trained end-to-end. We then present conformation generation experiment results of the proposed TensorVAE on two benchmark data-sets, GEOM-QM9 and GEOM-Drugs. These results are compared to those of **11** state-of-the-art baselines. In addition to conformation generation, we present

further experiment on molecular property prediction in Sec.A.7, where the performance of the proposed method is compared against **7** more state-of-the-art baselines on the MolecularNet benchmark (Wu et al., 2018).

## 3.1 EXPERIMENT SETUP

**Dataset.** Following existing work (Luo et al., 2021; Shi et al., 2021; Xu et al., 2021b;a; 2022; Zhou et al., 2022), we utilize the GEOM data-set for evaluating the performance of the proposed Tensor-VAE. GEOM contains 37 million energy and statistical weight annotated molecular conformations corresponding to 450,000 molecules (Axelrod & Gómez-Bombarelli, 2022). This dataset is further divided into two constituent datasets, Drugs and QM9. The Drugs dataset covers 317,000 median-sized molecules averaging 44.4 number of atoms. The QM9 dataset contains 133,000 smaller molecules averaging only 18 atoms. We randomly select 40000 molecules from each dataset to form the training set. For each molecule, we choose the top 5 most likely[2] conformations. This results in 200,000 training conformations for each train set. For validation set, we randomly sample 2,500 conformations for both Drugs and QM9 experiments. Finally, for testing, following (Shi et al., 2021; Xu et al., 2022), we randomly select 200 molecules each with more than 50 and less than 500 annotated conformations from QM9, and another 200 with more than 50 and less than 100 annotated conformations from Drugs[3].

**Determining input tensor graph size.** We conduct a basic data analysis on the entire Drugs dataset to determine the $98.5^{th}$ percentile of the number of atoms to be 69, and the percentage of molecules having more than 69 atoms and with more than 50 but less than 100 conformations is only $0.19\%$. Accordingly, we set the size of the input tensor to $69 \times 69$ for Drugs experiment. On the other hand, we use the maximum number of atoms 30 for QM9 experiment. The channel features for the input tensor include atom types, atom charge, atom chirality, bond type, bond stereo-chemistry and bond in-ring size. For the GDR tensor, we also include 3D coordinate channels and the computed distance channel. The resulting channel depth is 50 for GDR tensor and 46 for G tensor. The detailed information of these features and their encoding method is listed in Sec.A.4.

**Implementation details.** We implement the proposed TensorVAE using Tensorflow 2.3.1. All three transformer encoders of TensorVAE follow the standard Tensorflow implementation in `https://www.tensorflow.org/text/tutorials/transformer`. All of them have 4 layers, 8 heads and a latent dimension of 256. Both QM9 and Drugs experiments share the same network architecture and hyper-parameter configuration. We present the detailed training hyperparameter configuration in Sec.A.2.

**Evaluation metrics.** We adopt the widely accepted coverage score (COV) and matching score (MAT) (Shi et al., 2021) to evaluate the performance of the proposed TensorVAE model. These two scores are computed as.

$$\text{COV}\left(\mathbb{C}_g, \mathbb{C}_r\right) = \frac{1}{|\mathbb{C}_r|} \left| \left\{ R \in \mathbb{C}_r | \text{RMSD}\left(R, \hat{R}\right) \leq \delta, \forall \hat{R} \in \mathbb{C}_g \right\} \right| \tag{4}$$

$$\text{MAT}\left(\mathbb{C}_g, \mathbb{C}_r\right) = \frac{1}{|\mathbb{C}_r|} \sum_{R \in \mathbb{C}_r} \min \text{RMSD}\left(R, \hat{R}\right) \tag{5}$$

where $\mathbb{C}_g$ is the set of generated conformations and $\mathbb{C}_r$ is the corresponding reference set. The size of $\mathbb{C}_g$ is twice of that of $\mathbb{C}_r$, as for every molecule, we follow (Xu et al., 2022) to generate twice the number of conformations as that of reference conformations. $\delta$ is a predefined threshold and is set to 0.5Å for QM9 and 1.25Å for Drugs respectively (Shi et al., 2021) . RMSD stands for the root-mean-square deviation between $R$ and $\hat{R}$, and is computed using the `GetBestRMS` method in the `RDKit` (Riniker & Landrum, 2015) package. While COV score measures the ability of a model in generating diverse conformations to cover all reference conformations, MAT score measures how well the generated conformations match the ground-truth. A good generative model should have a high COV score and a low MAT score.

---

[2]Ranked by their Boltzmann weight.

[3]This limit on the number of conformations for testing molecules is taken directly from `https://github.com/DeepGraphLearning/ConfGF` which is also followed by all other compared methods.

**Baselines**. We compare the performance of the proposed TensorVAE model to those of 1 classical `RDKit` method; 5 distance-based methods including GraphDG, CGCF, ConfVAE, ConfGF and DGSM; 2 reconstruction-based methods including GeoDiff and Uni-Mol; 4 direct methods including CVGAE, GeoMol, DMCG and its symmetric permutation variant. The detailed information of the molecular property prediction baselines are presented in Sec.A.7

## 3.2 RESULTS AND DISCUSSION

The COV and MAT scores for all compared methods on both QM9 and Drugs datasets are presented in Tab.1. All experiments follow the same test data generation configuration in (Shi et al., 2021). Additionally, we have conducted three 3 ablation studies on the input feature engineering method in Sec.A.8 to demonstrate why 1D convolution with a $N \times 3$ kernel is necessary to achieve a good generative performance.

Table 1: Performance comparison between TensorVAE and 11 other SOTAs on GEOM dataset.

| Models | QM9 | | | | Drugs | | | |
|---|---|---|---|---|---|---|---|---|
| | COV (%) ↑ | | MAT (Å) ↓ | | COV (%) ↑ | | MAT (Å) ↓ | |
| | Mean | Median | Mean | Median | Mean | Median | Mean | Median |
| RDkit | 83.26 | 90.78 | 0.3447 | 0.2935 | 60.91 | 65.70 | 1.2026 | 1.1252 |
| CVGAE | 0.09 | 0.00 | 1.6713 | 1.6088 | 0.00 | 0.00 | 3.0702 | 2.9937 |
| GraphDG | 73.33 | 84.21 | 0.4245 | 0.3973 | 8.27 | 0.00 | 1.9722 | 1.9845 |
| CGCF | 78.05 | 82.48 | 0.4219 | 0.3900 | 53.96 | 57.06 | 1.2487 | 1.2247 |
| ConfVAE | 80.42 | 85.31 | 0.4066 | 0.3891 | 53.14 | 53.98 | 1.2392 | 1.2447 |
| ConfGF | 88.49 | 94.13 | 0.2673 | 0.2685 | 62.15 | 70.93 | 1.1629 | 1.1596 |
| GeoMol | 71.26 | 72.00 | 0.3731 | 0.3731 | 67.16 | 71.71 | 1.0875 | 1.0586 |
| DGSM | 91.49 | 95.92 | 0.2139 | 0.2137 | 78.73 | 94.39 | 1.0154 | 0.9980 |
| GeoDiff | 92.65 | 95.75 | 0.2016 | 0.2006 | 88.45 | 97.09 | 0.8651 | 0.8598 |
| DMCG | 94.98 | 98.47 | 0.2365 | 0.2312 | 91.27 | 100 | 0.8287 | 0.7908 |
| Uni-Mol | 97.95 | 100 | **0.1831** | **0.1659** | 91.91 | 100 | 0.7863 | 0.7794 |
| TensorVAE[1] | **98.11** ±0.25 | **100** ±0 | 0.1970 ±0.0016 | 0.1926 ±0.0027 | **94.91** ±0.35 | **100** ±0 | **0.7789** ±0.0027 | **0.7585** ±0.0076 |
| TensorVAE[2] | 97.11 ±0.31 | 100 ±0 | 0.2041 ±0.0046 | 0.1920 ±0.007 | 93.34 ±1.17 | 99.90 ±0.31 | 0.8074 ±0.0135 | 0.7927 ±0.0186 |

*Bold font indicates best result. Results for `RdKit`, CVGAE, GraphDG, CGCF, ConfGF are taken from (Shi et al., 2021); results for ConfVAE and GeoDiff are taken from (Xu et al., 2022); all other results are taken from (Zhou et al., 2022); TensorVAE[1] results and standard deviations are obtained by running 10 experiements each with a different random seed on a single 200 testing molecules set. TensorVAE[2] results and standard deviations are obtained by running 10 experiements each with a different random seed as well as a different set of 200 testing molecules.

In general, distance-based methods except for DGSM and ConfGF have relatively poor performance as compared to that of the classic RDKit. It has been argued that the the performance of `RDkit` is facilitated by an additional empirical force field (FF) (Halgren, 1996) optimization. Subsequently, CGCF and ConfVAE showed that with FF optimization, they can outperform `RDkit`. We also show the performance of the proposed TensorVAE with FF optimization compared to 5 other methods also employing FF optimization in Tab.3 in Sec.A.6, where TensorVAE outperforms all of them with a significant margin. However, this additional step further introduces complexity to the already complex two-stage generative model. ConfGF attempted to rectify this weakness by simulating a pseudo gradient-based force field. Such force field can be utilized in an annealed Langevin dynamics sampling to sequentially guide atom positions to a valid conformation. DGSM further improves the performance of ConfGF by using a dynamically constructed graph structure that is able to model long range atom interaction. Despite being posed as a direct method, both DGSM and ConfGF still need to compute atom distance as an intermediate step. Noticeably, DGSM also requires dynamically changing the graph structure for every sampling step. As shown in Tab.1, DGSM and ConGF outperform RDKit by a significant margin in both experiments. Nevertheless, their main weakness lies in the fact they they require numerous sampling steps to attain desirable performance. It has been reported in the ensuing work (Xu et al., 2022), that it took ConfGF approximately 8500 sec-

onds to fully decode 200 QM9 molecules and a staggering 11500 seconds for decoding 200 Drugs molecules. In constrast, **TensorVAE takes only 62 seconds using a single Xeon 8163 CPU core to decode 200 QM9 molecules, and 128 seconds for 200 Drug molecules**.

A main goal of GeoDiff is to be free of the dependence on distance. To achieve this, transformations are directly applied to 3D conformation, through which, a random or distorted conformation can be sequentially denoised to a valid conformation. An essential requirement for such transformation is that it needs to be SE(3) equivariant. This necessitates designing of sophisticated equivariant transition kernel. Despite its claim of not involving computing distance as an intermediate step, the equivariant graph field network (Satorras et al., 2021) used still relies on a distance-like quantity $\left\| x_i^L - x_j^L \right\|^2$ for every step of transformation. GeoDiff produces promising performance on both datasets. Unfortunately, it needs numerous diffusion steps ($T = 5000$) for generating conformations, requiring approximately the same amount of decoding time as ConfGF (Xu et al., 2022).

Uni-Mol circumvents this issue by reformulating molecular generation problem as an optimization problem. The structure of Uni-Mol is inspired by the Evoformer proposed in the Alphafold (Jumper et al., 2021) which also considers "atom-pair" interaction. This requires maintaining a pair interaction matrix ($N \times N$) through every attention layer. In addition, Uni-Mol has a significantly larger transformer structure as compared to ours, featuring 15 attention layers, 64 attention heads and a latent dimension size of 512. The Uni-Mol model is also first pretrained on a much larger dataset ($19e^6$ molecules, each with 10 conformations), and then fine-tuned on the GEOM dataset for conformation generation. Despite using a much smaller and also standard transformer model without pretraining, the proposed TensorVAE outperforms Uni-Mol on the Drugs dataset. Further, we show that the proposed model also performs competitively against Uni-Mol on a molecular property prediction task in Sec.A.7 without any pretraining, validating the effectiveness of the input feature engineering.

CVGAE is the first method proposed to generate molecular conformation directly from a 2D molecular graph. However, it yields the worst performance among the compared methods. DMCG attempted to revitalize the same framework by adapting a more advanced graph neural network structure combining GATv2 (Brody et al., 2021) with GN block (Battaglia et al., 2018). Noticeably, similar to Uni-Mol, it also maintains a $N \times N$ bond feature matrix throughout all layers. In constrast, we only use a standard transformer encoder architecture. Another major modification proposed in DMCG is introducing the permutation invariance of symmetric molecular substructures to the RMSD loss. Achieving this invariance requires enumerating all possible permutations which can become expensive for large molecules. The proposed TensorVAE outperforms DMCG with a significant margin. Finally, some samples of the TensorVAE generated conformations are shown in Sec.A.9

## 4 CONCLUSION

We develop TensorVAE, a simple yet powerful model that is able to generate 3D conformation directly from a 2D molecular graph. Unlike many existing work focusing on designing complex neural network structure, we focus on developing novel input feature engineering techniques. We decompose these techniques into three main ideas, and explain how one idea naturally evolves to the next. We first propose a tensor representation of a molecular graph. Then, we demonstrate that sliding a rectangle kernel through this tensor in an 1D convolution manner can achieve complete information aggregation. Finally, we present the complete CVAE-based framework featuring 2 transformer-based encoders and another transformer-based decoder, and propose a novel modification to the first multi-head attention layer of the decoder to enable sensible integration of the output of the other two encoders. We show through extensive experiments that with intuitive feature engineering, simple and standard model architecture can provide competitive performance compared to 18 recent state-of-the-art models. For future work, we plan to extend our method and methodology to tackling the challenging protein structure prediction.

## 5 REPRODUCIBILITY STATEMENT

We did not introduce nor have used any task-specific neural network archiecture. The results presented in this study can be straightforwardly reproduced using publically available datasets and ready-to-use implementation of convolution operation and Transformer from either `PyTorch` or `TensorFlow`. Specifically, we use the same transformer implementation found in `https://www.tensorflow.org/text/tutorials/transformer`. To compute RMSD, we first zero-center both the predicted and the ground-truth conformations. Then, we obtain the optimal rotation matrix using the Kabsch algorithm implementation found in `https://en.wikipedia.org/wiki/Kabsch_algorithm`. It is straightforward to implement this algorithm in a `python` function. Please note that if `TensorFlow` is used, this `python` function needs to be wrapped inside a `tf.py_function`, and then followed by `tf.stop_gradient` to prevent gradient update. Finally, we rotate the predicted conformation onto its ground-truth for calculating the RMSD. The above details together with the already presented hyper-parameter setting should suffice to reproduce our results.

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

# A APPENDIX

## A.1 GLOBAL INFORMATION AGGREGATION BEYOND THE $N^{th}$ HOP

A geometric interpretation of GNN's message passing layer is it aggregates information between atoms (and their bond) that are 1-hop away. With $L$ layers, information from atoms that are $L$-hop apart can be aggregated. Here, we define a global information aggregation as the $N^{th}$-hop aggregation with $N$ being the total number of atoms, where each atom is able to aggregate information from its farthest neighbour.

It is worth noting that for a fully-connected GNN, a 1-hop message passing can already achieve this global information aggregation. Transformer's self-attention can be considered as a type of fully-connected GNN. However, a vanilla transformer can only aggregate features from each token/atom; if edge features are not included, they needed to be incorporated somehow through additional inputs (e.g. the pair interaction matrix of Uni-Mol). The primary reason motivating the creation of the fully-connected tensor representation is we want each generated token contain both atom and bond features, such that we can eliminate the pair interaction or bond matrix. To achieve this, we fill each column of the fully-connected tensor with;

- focal atom features;
- chemical and virtual bond features indicating how the focal atom is connected to all other atoms;
- atom features of all connected atoms, since for each cell (except for cell of the focal atom) we sum atom features of both the connected atom and the focal atom.

In fact, running a $N \times 1$ kernel filter on the proposed tensor is conceptually similar to achieving a global information aggregation with a fully-connected GNN. By increasing kernel width to 3, the aggregation window also includes global information from two immediate neighbours. This type of information aggregation extends far beyond just $N^{th}$-hop.

More interestingly, when multiple kernels are applied simultaneously to the same $N \times 3 \times C$ region, each kernel is free to choose whichever group of atom/bond features to attend to depending on its kernel weights. This resembles the multi-head attention mechanism of a transformer, where each kernel(head) contributes to a portion of the generated feature token. We believe the effective global information aggregation driven by these two (tenor representation + 1D Conv) simple yet intuitive ideas is the main reason why the proposed TensorVAE achieves SOTA with much less number of parameters.

## A.2 TRAINING HYPERPARAMETERS

Training is conducted on a single Tesla V100 GPU. We follow a similar learning rate schedule, shown by Eq.3 of the original Transformer paper (Vaswani et al., 2017) but with $d_{model} = 9612$. This results in a maximum learning rate of $1.6e^{-4}$. To tackle the notorious issue of KL vanishing (Fu et al., 2019), we set a minimum KL weight of $1e^{-4}$ and double it every $62.5e^3$ iterations until a maximum weight of 0.0256 is reached. We select Adam optimizer (Kingma & Ba, 2015) for training. We present some interesting observations of the training/validation curve corresponding to this setup in Sec.A.5. For both experiments, the TensorVAE is trained for $1e^6$ iterations with a batch size of 128. The implementation details of NaiveUNet is explained in Sec.A.3

### A.3 NAIVEUNET MODEL ARCHITECTURE

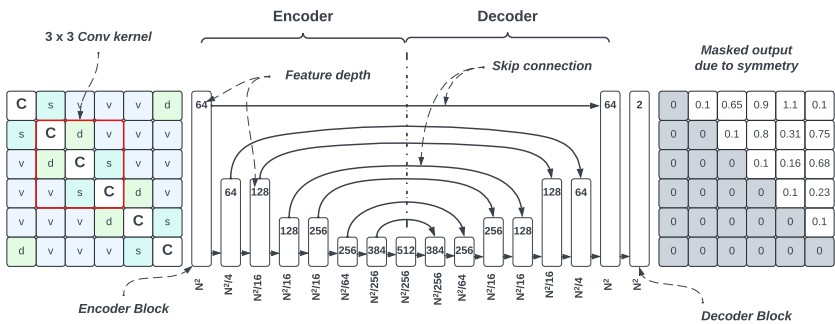

Figure 4: Naive UNet model. $N = 69$

We train the above NaiveUNet on the Drugs dataset for 30 epochs with a constant learning rate of $1e^{-4}$, and batch size of 32. We follow the same method presented in GraphDG (Simm & Hernández-Lobato, 2019) to convert the predicted distance matrix to conformation.

### A.4 ATOM AND BOND FEATURES

We list the atom features and bond features together with the encoding method used to construct the proposed tensor in Tab.2.

Table 2: Atom and bond features used to construct input tensor.

| Feature name | Feature value | Encoding method |
|---|---|---|
| Atom type | H, C, N, O, F, S, Cl, Br, P, I, Na, B, Si, Se, K, Bi | one-hot |
| Atom charge | -2, -1, 0, 1, 2, 3 | one-hot |
| Atom chirality | Unspecified, Tetrahedral_CW Tetrahedral_CCW, Other | one-hot |
| Bond type | Single, Double, Triple, Aromatic, Virtual | one-hot |
| Normalized bond length | - | real-value |
| Bond stereochem | StereoNone, StereoAny, StereoZ StereoE, StereoCIS, StereoTrans | one-hot |
| Bond in-ring size | 3 - 10 | one-hot |
| Coordinate (3 channels) | - | real-value |
| Pair wise atom distance | - | real-value |

### A.5 TRAINING AND VALIDATION CURVE

We present the train and validation plots for KL and reconstruction loss based on Drugs dataset in Fig.5a and Fig.5b, respectively. Both plots are based on an initial KL weight of $1e^{-4}$ doubling every $62.5k$ iterations (40 epochs). While KL validation loss reached 18.29 after $1e^6$ iterations (640 epochs), the reconstruction/RMSD loss reached $0.64$Å at the end of training. During the first 5 epochs of training, model learning focused on reducing the KL loss due to it is orders of magnitude larger than the RMSD loss. We were expecting this trend to continue for a while until both losses converge roughly in the same range. However, much to our surprise, the model seemed to find a way to drastically reduce RMSD loss much earlier by leveraging the information from the GDR encoder; it learned to "cheat" by directly reversing coordinate information embedded in the output of GDR encoder back to the original conformation. The RMSD loss dropped to as low as $0.08$Å. On the other hand, the KL loss climbed to almost 800, signaling signifcant divergence from standard normal distribution. At this stage, output of the GDR encoder contains informative features of the original 3D coordinates. With the KL loss weight increasing, it becomes more difficult for the model to cheat since training is forcing the output of GDR encoder to conform to a standard uninformative Gaussian distribution. The KL loss started to drop while the RMSD loss remained steady, indicating increasing reliance on the output of G encoder for reconstructing the conformation. As the output of GDR encoder becomes less informative, the model learned to rely almost entirely on the aggregated feature from the G encoder to decode conformation.

We attempted to initiate the training with a much larger initial KL weight ($1e^{-2}$) to prevent "cheating" from begining. However, this quickly led to the notorious KL vanishing issue (Fu et al., 2019). We figure that "cheating" is actually beneficial in that it reduces learning difficulty particularly for the decoder; its weights are tuned on easy training task, simply reversing what GDR encoder has done. In other words, the tuned weights of the decoder already hold crucial information on how to decode highly informative input features. As KL weight increases, model learning shifts to make the output of G encoder more informative. Also, this maybe an easier learning task as the RMSD loss is already very low (back-propagation of this loss contributes little to weight update); instead, model learning primarily focuses on optimizing the KL loss. This two-stage iterative loss optimization is much easier than optimizing both losses simultaneously throughout the training process.

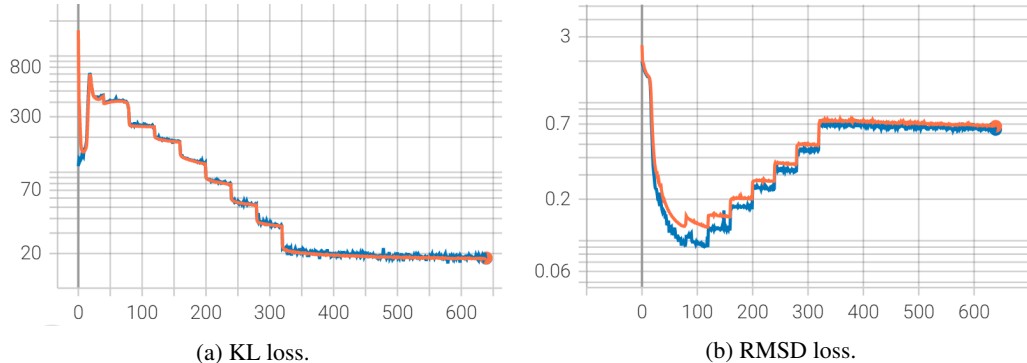

(a) KL loss.  (b) RMSD loss.

Figure 5: Training and validation plots for Drugs dataset. Orange line: Train; Blue line: Validation

## A.6 GENERATION PERFORMANCE ON DRUGS DATASET WITH FORCE FIELD OPTIMIZATION

Table 3: Performance comparison between methods with FF optimization.

| Method | COV | | MAT | |
|---|---|---|---|---|
| | Mean | Median | Mean | Median |
| CVGAE + FF | 83.08 | 95.21 | 0.9829 | 0.9177 |
| GraphDG + FF | 84.68 | 93.94 | 0.9129 | 0.9090 |
| CGCF + FF | 92.28 | 98.15 | 0.7740 | 0.7338 |
| ConfVAE + FF | 91.88 | 100 | 0.7634 | 0.7312 |
| GeoDiff + FF | 92.27 | 100 | 0.7618 | 0.7340 |
| TensorVAE + FF | **96.15** | **100** | **0.6723** | **0.6605** |
| | ±0.34 | ±0 | ±0.0023 | ±0.0064 |

*Results for CVGAE, GraghDG, CGCF, ConfVAE and GeoDiff are taken from (Xu et al., 2021b); Standard deviations are obtained by repeating experiments 10 times each with a different random seed on a test set with 14396 conformations.

## A.7 MOELCULAR PROPERTY PREDICTION RESULTS

We conduct further experiment on molecular property prediction to demonstrate the effectiveness of the proposed feature engineering method. Following Uni-Mol (Zhou et al., 2022) and GEM (Fang et al., 2022), we report property prediction result on the MolecularNet QM9 regression task (https://moleculenet.org/datasets-1). The goal of this task is to estimate *homo*, *lumo*, and *homo-lumo gap* properties of molecules in the QM9 dataset based on their molecular structure. This task is different from the conformation generation experiment in that both 3D coordinates and 2D molecular graph of a molecule are supplied as input to a prediction model. The accuarcy of property prediction depends on how well a model can extract and aggregate such input information among atoms.

Similar to Uni-Mol, we adapt the proposed GDR encoder to this regression task by changing its prediction head. Specifically, we use the same GDR transformer encoder structure as presented in Fig.3 (with only 4 attention layers) and add an additional mean pooling layer, which is then followed by a linear layer for property prediction. To obtain training data, we follow the same data train-val-test split[4] in Uni-Mol and GEM and standardize the output property data. This results in 106362 train samples, 13299 val samples and 13356 test samples. We train the adapted model for 300 epochs with a batch size of 128. The learning rate schedule is the same as TensorVAE. We report the mean average error(MAE) over all the test samples.

The result of the adapted model is compared to those of 7 other models including;

- D-MPNN (Yang et al., 2019), AttentiveFP (Xiong et al., 2019) and GEM which are GNN based models without pretraining;
- N-Gram (Liu et al., 2019), PretrainingGNN (Hu et al., 2019) and GROVER (Rong et al., 2020) with pretraining. In particular, GROVER integrates GNN into a Transformer architecture, and there are two variants with different model capacity, $GROVER_{base}$ and $GROVER_{large}$;
- and finally, three variants of Uni-Mol including one without pretraining, another without using the $N \times N$ pair representation matrix, and the complete version.

The MAE for all compared methods are summaried in Tab.4. The proposed method produces a competitive performance, only underperforming the complete Uni-Mol setup with pretraining and also considering pair representation. This experiment demonstrates that the proposed feature engineering method is very effective at global information aggregation.

Table 4: Property prediction result comparison based on MolecularNet QM9 benchmark.

| Method | MAE |
| --- | --- |
| D-MPNN | 0.00814 (0.00001) |
| AttentiveFP | 0.00812 (0.00001) |
| N-Gram | 0.00964 (0.00031) |
| PretrainGNN | 0.00922 (0.00004) |
| GROVER base | 0.00984 (0.00055) |
| GROVER large | 0.00986 (0.00025) |
| GEM | 0.00746 (0.00001) |
| Uni-Mol w/o pair representation | 0.00573 (0.00004) |
| Uni-Mol w/o pretraining | 0.00653 (0.00040) |
| Uni-Mol | **0.00467** (0.00004) |
| GDR encoder (ours) | 0.00553 (0.00012) |

*All results are taken from (Zhou et al., 2022). Values in parenthesis are standard deviation obtained by repeating experiments 4 times.

---

[4]The data is split with a ratio of $8 : 1 : 1$ using scaffold spliting while considering chirality

A.8    ABLATION STUDIES

In this section, we further demonstrate the effectiveness and necessity of running an 1D convolution with $N \times 3$ kernels over the proposed input tensor through 3 ablation studies.

**Why is 1D convolution necessary**. We have shown a model based on a $3 \times 3$ kernel in Sec.A.3 called NaiveUNet. Here, we provide a more detailed analysis of why NaiveUNet produces unsatisfactory result. The primary reason for this poor performance is the "field of view" of a conventional $d \times d$ $(d < N)$ kernel only sees a partial connection pattern of a focal atom. In comparison, a $N \times 3$ kernel's "field of view" encompasses the complete connection pattern of a focal atom. We further observe that when applying a $3 \times 3$ kernel filter to the top left region of the proposed tensor, its field of view only includes a focal atom, its two neighboring atoms and how the focal atom is connected to them. There are two main disadvantages associated with this. Firstly, it only achieves a 1-hop information aggregation. Secondly when the $3 \times 3$ kernel moves to an off-diagonal part of the tensor, where most connections are virtual bonds (as atoms of a molecule are often sparsely connected), information aggregation occurs mostly between atoms that are not chemically connected and is therefore less meaningful than that on the diagonal part of the tensor. For these two reasons, the NaiveUNet's performance on the GEOM Drugs dataset is the worst as shown in Tab.5.

**What happens if we remove all virtual bonds**. Notice that if we remove all the virtual bonds in each column and still run a $N \times 3$ kernel through the tensor, its "field of view" is a "2-hop atomic-environment" (because the focal atom can "see" how neighboring atoms are chemically connected to all their direct neighbors). Another observation is that after removing all virtual bonds, each column does not correspond to a fully-connected GNN. Therefore it no longer enables a global information aggregation. The conformation generation results of this variant of TensorVAE on Drugs dataset is shown as as TensorVAE abla1 in table below. It is observed that due to local-only information aggregation as a result of removing all virtual bonds (and related atom features), the performance is worse than the complete TensorVAE version.

**What happens if a $N \times 1$ kernel is used**. The final ablation study concerns with using a $N \times 1$ kernel with a smaller "field of view" as compared to that of a $N \times 3$ kernel. Its performance on Drugs dataset is shown as TensorVAE abla2 in Tab.5. It performs slightly better than the ablation removing all virtual bonds. The reason is that though its field of view is smaller, it still achieves a global information aggregation for the focal atom. Nevertheless, it underperforms the complete TensorVAE version due to a smaller "field of view" for information aggregation.

Table 5: Performance comparison among models with different input feature engineering setupon GEOM Drugs dataset

| Method | COV | | MAT | |
| --- | --- | --- | --- | --- |
| | Mean | Median | Mean | Median |
| NaiveUNet | $52.14 \pm 1.48$ | $51.69 \pm 1.17$ | $1.4322 \pm 0.0247$ | $1.3861 \pm 0.0173$ |
| TensoVAE abla1 | $90.72 \pm 1.54$ | $99.53 \pm 0.64$ | $0.8748 \pm 0.0161$ | $0.8619 \pm 0.0214$ |
| TensoVAE abla2 | $91.04 \pm 1.21$ | $99.74 \pm 0.42$ | $0.8706 \pm 0.0131$ | $0.8561 \pm 0.0204$ |
| TensorVAE | $\mathbf{93.34} \pm 0.35$ | $\mathbf{99.90} \pm 0.31$ | $\mathbf{0.8074} \pm 0.0135$ | $\mathbf{0.7927} \pm 0.0186$ |

*The standard deviations for all ablation studies are obtained by testing on 2000 testing molecules.

## A.9    SAMPLES OF GENERATED CONFORMATIONS

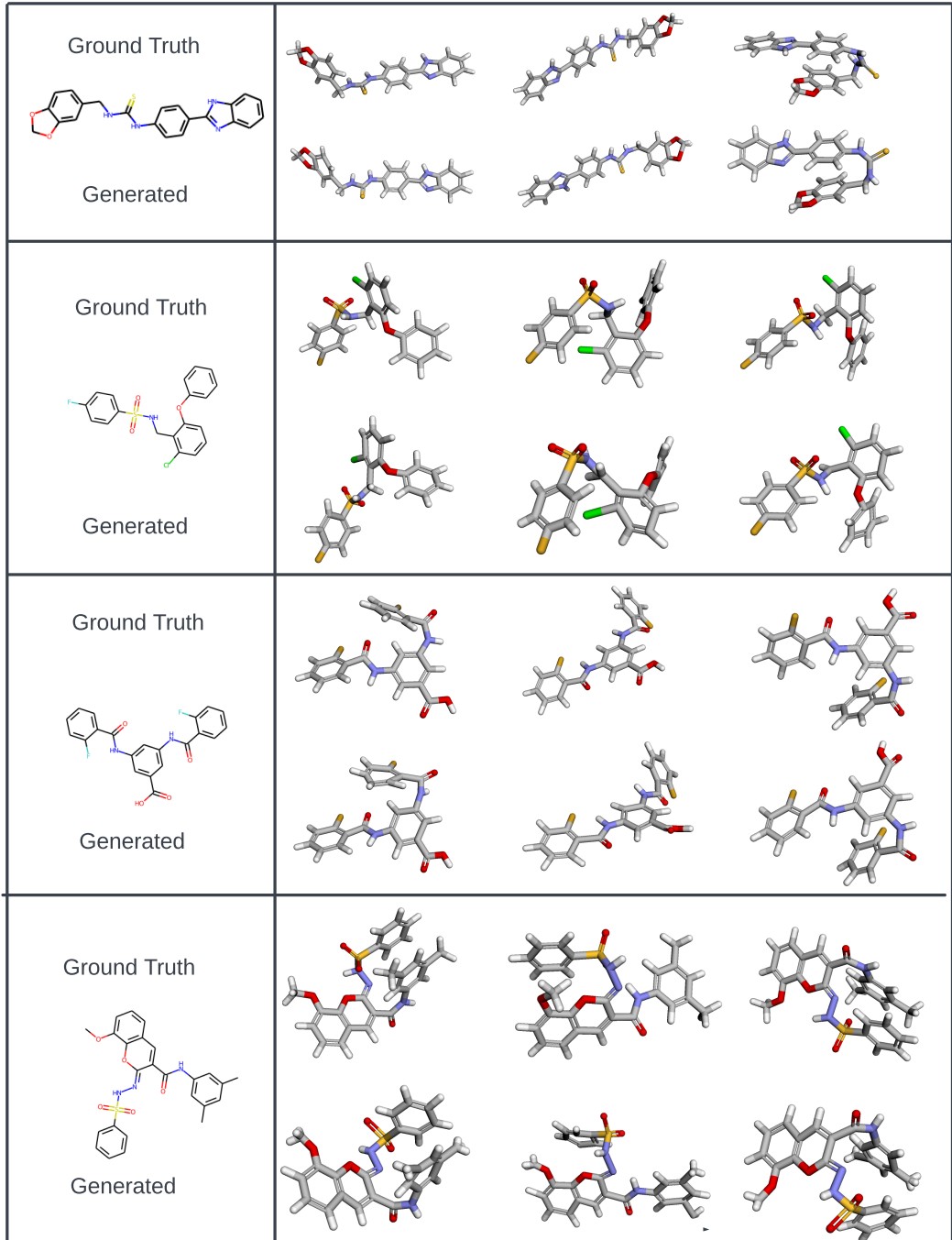

Figure 6: Generated samples by the TensorVAE

