# OpenReview forum: "TensorVAE: A Direct Generative Model for Molecular Conformation Generation driven by Novel Feature Engineering"
_ICLR.cc/2023/Conference — Submitted to ICLR 2023_

### Official Review · Reviewer_8zkT · 2022-10-18

**Confidence:** 4
**Correctness:** 3
**Technical Novelty And Significance:** 2
**Empirical Novelty And Significance:** 2
**Recommendation:** 5

**Clarity, Quality, Novelty And Reproducibility:**

The paper is fairly easy to follow, and the main ideas are presented in sufficient detail.

There are several typos and grammatical errors etc. throughout the paper, which are easily fixed (not important for my assessment).

The definition of COV in eq. 4 seems to be incorrect (should not be for all R hat in C_g)

There appears to be some typos in the results in table 1 (at least in comparison with the corresponding table in the Uni-Mol paper.)

**Strength And Weaknesses:**

Strengths:
The paper deals with an important problem and presents a working solution that is fairly simple and effective.

Weaknesses:
The novelty is limited - the main technical contribution appears to be a model that works well, built using standard CNN, transformer, VAE components.

It is unclear if the comparison with other methods is fair - results are taken directly from previous papers but is it the same test set that is used? According to the text, the authors randomly select 200 molecules.

Confidence intervals are not provided for the results. With such a small test set, significance is difficult to determine.

Because a CNN is used, it appears that the method is not invariant to permutation of the atoms. It is not clear if this has negative implications for the performance.

It is not clear to me what the motivation is behind combining node and edge features in a tensor. This restricts the dimensions of the node and edge features to be identical, but on the other hand allows for standard CNN code to run on the representation. Is that the reason for this choice?

The relation of the proposed model to common GNN architectures is not fully described. If I understand correctly, a kernel of size 3xN is used? If it had been a 1xN kernel, I believe it would correspond exactly to a fully connected GNN which is commonly used.


**Summary Of The Paper:**

The paper presents a CNN/transformer based variational autoencoder for generation of molecular 3d structure from molecular graphs.


**Summary Of The Review:**

While I find the subject of the paper very interesting, my primary reason for recommending rejection is the lack of novelty and the limited empirical evaluation.

---

> ### Author Response · Authors · 2022-11-16
> **Response to reviewer 8zkT**
>
> We thank reviewer 8zkT again for pointing out the results difference between ours and Uni-Mol's, which help confirm the SOTA status of the proposed TensorVAE. We have striven to address most of your major concerns in the general responses, particularly regarding lack of novelty and limited empirical evaluation. Here, we provide additional responses to 5 specific comments.
>
> ### It is unclear if the comparison with other methods is fair - results are taken directly from previous papers but is it the same test set that is used? According to the text, the authors randomly select 200 molecules. Confidence intervals are not provided for the results. With such a small test set, significance is difficult to determine.
>
> Thank you very much for the comment. Please refer to our general response 1 for how we have addressed these issues.
>
> ### Because a CNN is used, it appears that the method is not invariant to permutation of the atoms. It is not clear if this has negative implications for the performance.
>
> Thank you for your keen observation. After further investigation, we discover that running a 1D convolution over the proposed tensor preserves the ordering of atoms but it does not preserve the values of the feature tokens. The feature token generated for a focal atom in the bottom right corner is different from that generated when the focal atom is shuffled to the top left corner. This is because, although the connection pattern remains the same, the kernel weights attending to these patterns are not the same due to the change of focal atom location. Therefore, this feature engineering is not permutation invariant. We have removed this claim in the revised submission.
>
> When deciding the ordering of the atoms for constructing the input tensor, we have followed [GraphDG](https://github.com/gncs/graphdg/blob/a138448de24fcc1ead3284e380ace5e88a2c6643/graphdg/parse/extended_graph.py#L12) to traverse the molecular graph with a random breadth-first-search (BFS) method. This leads to a complete random ordering of atoms for every training sample. Also note that we collect 5 graph-conformation pairs from each molecule to form the training dataset. The ordering of atoms in each pair corresponding to the same molecule is different as a result of the random BFS. We are still able to achieve SOTA performance with such random permutation of the atom ordering, proving there is no negative implication.
>
> ### It is not clear to me what the motivation is behind combining node and edge features in a tensor. This restricts the dimensions of the node and edge features to be identical, but on the other hand allows for standard CNN code to run on the representation. Is that the reason for this choice?
>
> Thanks for the excellent question. Stacking node and edge features in a single feature vector and extending this vector across all cells of the tensor enables a 1D convolution to aggregate all connected atom and bond features onto the focal atom in a single operation. This eliminates the need of maintaining another interaction/bond matrix such as that in DMCG or Uni-Mol for effective information aggregation. It also leads to a significant reduction of model parameters.
>
> ### The relation of the proposed model to common GNN architectures is not fully described. If I understand correctly, a kernel of size 3xN is used? If it had been a 1xN kernel, I believe it would correspond exactly to a fully connected GNN which is commonly used.
>
> Thank you so much for your comment. We have addressed this comment in the general response 3. On a different note, among all the compared methods, only Uni-Mol and the proposed TensorVAE use a transformer backbone which corresponds to a fully-connected GNN. All other compared methods use regular sparsely-connected GNN.
>
> ### The definition of COV in eq. 4 seems to be incorrect (should not be for all $\hat{R}$ in $C_g$)
>
> Thanks for the comment. The definition of COV is the percentage of the conformations in the reference set that are matched by at least one conformation in the generated set with a RMSD value less than $\delta$. So for each reference conformation, we need to calculate RMSDs over the entire set of generated conformations and determine the minimum RMSD. This is the same COV definition and equation that have been used in all the compared baselines.

---

> > ### Comment · Reviewer_8zkT · 2022-12-05
> > **Thank you**
> >
> >
> > Thank you for the detailed explanations and additional results. The added ablation study, especially the 1xN convolution case, is interesting and gives a better understanding of the proposed method. It is very encouraging to see that the method achieves its good results with a relatively small model size. I appreciate the effort to include standard deviations for the results in Table 1.
> >
> > I still think there are some issues, that should be cleared up, which would make the paper substantially better.
> > * As you mention in the rebuttal, the method is not permutation invariant (which one might think would be a desirable property) but it seems to work well anyway. The reason for the lack of invariance is the 3xN kernel - with a 1xN kernel the model is invariant, but performance is a lot worse. This, I do not completely understand, but maybe a model with 1xN convolutions but more layers or larger embeddings would be better? A better argumentation for this choice or a more convincing empirical demonstration would be enlightening.
> > * It is not so clear to me how how important the "feature engineering", highlighted as a central contribution, is the the method. The individual features used to describe the atoms and their local properties as well as bonds etc. appear to be fairly standard, following what has been done also in the "non neural network" literature. I believe you are correct that the choice of features are important, but the novelty and significance of the particular choices made is not absolutely clear to me, from the paper.
> >
> > Based on this, I have improved my score slightly.
> >
> >
> > Minor point (not important for assessment)
> > The definition of COV in eq. 4 seems to be incorrect (should not be for all \hat R in C_g)
> > * Thanks for the comment. The definition of COV is the percentage of the conformations in the reference set that are matched by at least one conformation in the generated set with a RMSD value less than. So for each reference conformation, we need to calculate RMSDs over the entire set of generated conformations and determine the minimum RMSD. This is the same COV definition and equation that have been used in all the compared baselines.
> >
> > I believe you compute this correctly - my point was just that the "for all" symbol should not be there in the mathematical definition of coverage (like it is defined in Shi et al.)

---

> > > ### Author Response · Authors · 2022-12-06
> > > **Response to questions 2 & 3**
> > >
> > > ### It is not so clear to me how how important the "feature engineering", highlighted as a central contribution, is the the method. The individual features used to describe the atoms and their local properties as well as bonds etc. appear to be fairly standard, following what has been done also in the "non neural network" literature.
> > >
> > > Thanks for the comment. We agree with your observation that we have used a set of standard atom and bond features common to other compared methods. In fact, this is done on purpose to eliminate as much as possible other unrelated factors clouding the importance of the proposed feature engineering. We'd like to emphasize the main contribution here again. We develop a feature engineering method that takes **standard atom and bond features** as input and transforms these features into token-like feature vectors that are consumed by a **standard (and a much smaller) transformer backbone**. This promising performance of the proposed TensorVAE does not come from any special molecular features nor from any specialized model architecture; it is solely attributed to the proposed feature engineering. Having established the feature engineering as the main contributing factor of the performance, we further elaborate on the rationale of the feature engineering design in three main ideas which are also the main ideas that our work resolves around.
> > >
> > > 1. **Tensor representation of molecular features**. The main purpose of the tensor representation is to combine atom features and bond features as a single input. As each cell of the tensor is a vector stacked with both atom and bond features, the feature token generated by a 1D convolution operation contains both these features. We choose this input representation as it eliminates the need to maintain pair interaction or bond matrix as in Uni-Mol and DMCG. This leads to a significant reduction of model parameters required to achieve SOTA performance.
> > >
> > > 2. **Extended kernel size**. Running a conventional $3 \times 3$ kernel on the proposed tensor representation does not exploit adequately the fully connected nature of the input tensor. It sees only partial connection pattern of an atom most of the time, yielding poor performance as shown in ablation study 1. We choose the extended kernel size to enable a global information aggregation in each stride of the convolution. By extending kernel width to 3, each stride also aggregates global information for two neighboring atoms. More importantly, with multiple kernel attending to the same region, the feature aggregation resembles the multi-head attention of a transformer. Due to our design choice, all of this information aggregation is achieved in a single operation. In comparison, this is impossible to achieve with a single layer-operation of a GNN.
> > >
> > > 3. **Modified attention**. We have also carefully studied the effect of different integration method between the output of the G encoder and the GDR encoder, including stacking and addition that are commonly used in other VAE-based works. Both methods yielded poor results in our case (RMSD and KL losses were not decreasing for many epochs, so we gave up on them quickly). After careful consideration, we choose to compute Query and Key matrices from the output of the G encoder and compute Value matrices from the output of the GDR encoder. This is because Query and Key matrices are used to compute attention weights that dictate how much influence an atom exerts on the other. We hypothesize this influence should be mainly driven by the 2D molecular structure and stay stable for the same molecular graph for effective information aggregation. Subsequently, we attribute the variation in conformation to the randomness of the Value matrices. This is a sensible choice as the Value matrices coming from the output of GDR encoder are pushed to be Gaussian distributed via KL loss minimization during training. With this design choice, the training of TensorVAE is stable as shown in Figure 5 of the manuscript. The final performance also attains SOTA.
> > >
> > > Each of the main ideas serves an indispensable function to push the performance of the final model. They are tightly coupled and carefully designed to fully realize the potential of a simple and standard model to achieve a superior performance.
> > >
> > > ### I believe you compute this correctly - my point was just that the "for all" symbol should not be there in the mathematical definition of coverage (like it is defined in Shi et al.)
> > >
> > > Thanks for the comment. We will update Eq.(4) in the revised version of the manuscript accordingly.

---

> > > ### Author Response · Authors · 2022-12-06
> > > **Response to question 1**
> > >
> > > Reviewer 8zkT' comments continue to enlighten us and provide further insight into the proposed method. Again, we are extremely grateful for your feedback. We provide detailed response to your further comments below.
> > >
> > > ### The reason for the lack of invariance is the 3xN kernel - with a 1xN kernel the model is invariant, but performance is a lot worse. This, I do not completely understand, but maybe a model with 1xN convolutions but more layers or larger embeddings would be better? A better argumentation for this choice or a more convincing empirical demonstration would be enlightening.
> > >
> > > Thanks for the insightful comment. **The convolution with a $N\times 1$ kernel is not permutation invariant either**. Recall that we represent each atom of a molecule with a cell on the diagonal of the proposed input tensor. We refer to atom features placed on diagonal as the focal atom features. As a result, shuffling the ordering of atoms of a molecule not only **changes the ordering of focal atom feature along the diagonal of the input tensor** but also **changes their cell positions in each column**. For example, when the “first” atom of a molecule is permuted to be the “last” atom, the corresponding position of the focal atom feature changes from the first cell in the first column to the last cell of the last column of the input tensor. Consequently, the ordering of other connected atoms (and their associated atom and bond features) in each column also shifts accordingly. **Although the connection pattern remains the same in each column, the positions of the cells are changed due to permutation**. Meanwhile, the row-wise ordering of the kernel weights for a $N\times 1$ kernel remains the same for all columns of the tensor. This leads to different kernel weights attending to the same focal atom features for each random permutation. Therefore, the resulting feature token values generated for each atom are also different for each random permutation. **In fact, this is exactly the same reason for why convolution with a $N\times 3$ kernel is not permutation invariant** as we have explained in our previous response. So for both kernel sizes (and for all other kernel sizes), the convolution operation is not permutation invariant.
> > >
> > > As per reviewer 8zkT's previous enlightening comment, convolution with a $N \times 1$ kernel can be considered as a fully-connected GNN. Feeding feature tokens generated from a GNN to a transformer backbone has already been well explored in the [GROVER](https://arxiv.org/abs/2007.02835) model. As shown in Table 4 of Sec A.7 of the revised manuscript, the performance of the GROVER model is much worse as compared to the proposed TensorVAE (the GDR encoder part) despite having a significantly larger model capacity(**100M model parameters**). This combined with our ablation study 3 clearly shows that the information aggregation enabled by a GNN (or $N \times 1$ kernel in our case) is not as effective as that achieved by a $N \times 3$ kernel.

---

> > > > ### Comment · Reviewer_8zkT · 2022-12-07
> > > > **Thank you**
> > > >
> > > > Yes, I see: The model is not invariant even with an 1xN kernel. I just cannot help wonder if this is a suboptimal design choice. I appreciate that this representation allows a standard CNN architecture, but in my experience building in the known invariances in the architecture usually is a large benefit. On the other hand, your proposed architecture in a way forces the role of the node and edge features to be the same - maybe that is somehow an advantage. I think it would really strengthen the paper to make a direct comparison with a fully connected GNN that otherwise mimics your setup.

---

> > > > > ### Author Response · Authors · 2022-12-09
> > > > > **Additional experimental results with fully connected GNN**
> > > > >
> > > > > We conduct further experiment on a variant of TensorVAE employing a $1 \times 1$ convolution operation. This setup corresponds exactly to connecting a fully-connected GNN with a standard transformer backbone for conformation generation. We experimented with 6 hyper-parameter configurations listed as following. All experiments were conducted on GEOM Drugs dataset.
> > > > >
> > > > > | Model name| Embedding size| KL weight schedule|No. of transformer layers|No. of parameters|
> > > > > | -----------| ----------- | ----------- | ----------- |----------- |
> > > > > |GNN_vanila|256|same as TensorVAE|same as TensorVAE(4)|6.5M|
> > > > > |GNN_large_1|320|same as TensorVAE|same as TensorVAE(4)|11M|
> > > > > |GNN_large_2|256|same as TensorVAE|6|10M|
> > > > > |GNN_large_3|320|1e-5 doubling every 16 epochs|4|11M|
> > > > > |GNN_large_4|320|1e-6 doubling every 16 epochs|4|11M|
> > > > > |GNN_large_5|320|1e-7 doubling every 16 epochs|4|11M|
> > > > >
> > > > > GNN_vanlia has the same hyper-parameter configuration as TensorVAE except that it employs a smaller $1 \times 1$ kernel. Consequently, its number of parameters is 6.5 M which is significantly less than that of TensorVAE. There are two ways to increase the number of parameters of the GNN-based variant to match that of the TensorVAE employing a $N \times 3$ kernel for a fair comparison, including a larger embedding size and more transformer layers. These two setups correspond to GNN_large_1 and GNN_large 2. Unfortunately, training for these 3 setups failed to reduce RMSD error after more than 10 epochs of training; we kept facing the KL vanishing problem. To tackle this, we experimented with 3 more configurations (GNN_large_3,4 and 5) with much lower KL weights and shorter step period to force training to focus more on reducing the RMSD loss. Unfortunately again, after more than 40 epochs (25+ hours) of training all three efforts have also failed to resolve this issue. **We have attached the training and validation curve for all 6 experiments [here](https://imgur.com/BxgnUZo)**. It is observed that for all cases, while KL error quickly decreases to close to zero, the RMSD loss stays almost constant at 4.0, indicating model's inability to learn. We will also release code to reproduce the above after double blind review ends.
> > > > >
> > > > > It seems that the GNN-based model is struggling to learn any meaningful information that contribute to producing valid conformations. Instead, it always resorts to reducing KL loss which is a much easier learning task. **This fact combined with our ablation study 3 manifest an emerging trend that the TensorVAE model's capacity to learn difficult conformation generation task improves with the increase of expressive power of its aggregation mechanism**. In other words, the extra flexibility introduced by the increased kernel size (from $1 \times 1$ to $N \times 3$) is the main contributing factor to the promising performance of the TensorVAE model. On the other hand, the impact of permutation invariance property is minimal **in our specific case**. Therefore, **we conclude that the design choice made to use a $N \times 3$ kernel is sensible and fully justified**.

---

> > > > > ### Author Response · Authors · 2022-12-09
> > > > > **A unified framework**
> > > > >
> > > > > Thanks for the comment. We will address your concern in two parts. In the first part, we show that a fully-connected GNN is a special case equivalent to running a $1 \times 1$ convolution over the proposed tensor representation. In the second part, we show additional experimental results of adopting a fully-connected GNN into the proposed framework.
> > > > >
> > > > > The main reason for the permutation invariance property of a GNN is attributed to the sharing of weight matrices among nodes and edges. As node or edge features are multiplied by the same weight matrices, any permutation of their ordering yields the same aggregated feature values. We can also achieve this invariance through a $1 \times 1$ convolution operation over the proposed fully-connected tensor. With a $1 \times 1 \times F \times C$ kernel size (where F is the number of kernels), a weight matrix $W \in R^ {F \times C}$ is shared among all $N\times N$ cells of the tensor. ***Since each cell, regardless it is on-diagonal or off-diagonal, is stacked with an atom feature vector and a bond feature vector, the weight matrix can be decomposed into two parts***, $W_v \in R^{F \times C_v}$ and $W_e \in R^{F \times C_e}$, where $C_v$ is the atom feature vector size and $C_e$ is the bond feature vector size. The bond feature vector for on-diagonal cells is filled with zeros, since there is no self connection for focal atoms. Virtual/chemical bond only exists between different atoms. Please also note that, ***we never restrict the dimension of an atom feature vector to be the same as that of a bond feature vector, these two feature vectors have different dimension and are concatenated in each cell***.  Subsequently, for each column $n$ of the tensor, a $1 \times 1$ convolution operation followed by a sum-aggregation over the rows can be reduced to the following equation;
> > > > >
> > > > > $$h_n = \mathbf{ReLu}\left(W_vh_n^0 + \sum_{m\in N_{\setminus n}}\left(W_vh_m^0 + W_eh_{nm}\right)\right) = \mathbf{ReLu}\left(W_vh_n^0 + \sum_{m\in N_{\setminus n}}W\mathbf{concat}(h_m^0,h_{nm})\right)$$
> > > > >
> > > > > where $h_n\in R^{F \times 1}$ is the feature token generated for the $n^{th}$ atom and $h_n^0 \in R^{C_v\times 1}$ is the focal atom feature on-diagonal. $h_{m}^0 \in R^{C_v\times 1}$ and $h_{nm} \in R^{C_e\times 1}$ are the connected atom feature and bond feature off-diagonal, respectively. **Noticeably, the above equation corresponds exactly to the canonical information aggregation of a fully-connected GNN (to be precise, a [GraghSage](https://cs.stanford.edu/people/jure/pubs/graphsage-nips17.pdf) GNN with a sum aggregation)**. While all atom feature vectors share the same weight matrix $W_v$, all bond feature vectors share another matrix $W_e$. As a result, the feature token generated is also permutation invariant. Additionally, as atom features and bond features are weighted differently, **their roles are not the same in terms of contributing to the aggregated feature values**.
> > > > >
> > > > > Similarly, the feature aggregation operation of a $N \times 1$ kernel can be expressed as;
> > > > >
> > > > > $$h_n = \mathbf{ReLu}\left(W_v^nh_n^0 + \sum_{m\in N_{\setminus n}}\left(W_v^mh_m^0 + W_e^mh_{nm}\right)\right)$$
> > > > >
> > > > > Notice that there is a different matrix ($W_v^m$ and $W_e^m$) for each row of the column which invalidates the permutation invariance of the aggregation. However, this type of aggregation is more flexible and has more expressive power as different node and edge features are weighted differently. This flexibility is further increased with a $N \times 3$ kernel whose corresponding aggregation can be expressed as;
> > > > >
> > > > > $$h_j = \mathbf{ReLu}\left(\sum_c^{(i,j,k)} W_v^ch_c^0 + \sum_c^{(i,j,k)} \sum_{m}^{N_{\setminus c}}W_v^mh_m^0 + W_e^mh_{nm}\right)$$
> > > > >
> > > > > where $i,j,k$ are the indices of three adjacent columns. In this respect, **the information aggregation achieved by a fully-connected GNN is a special case (the simplest form) of a more general framework embodied by a single convolution operation over the proposed tensor representation**.

---

### Official Review · Reviewer_4X5j · 2022-10-21

**Confidence:** 5
**Correctness:** 4
**Technical Novelty And Significance:** 2
**Empirical Novelty And Significance:** 2
**Recommendation:** 5

**Clarity, Quality, Novelty And Reproducibility:**

The overall flow of the paper is quite good, and the explanations and figures and diagrams are very much appreciated. The authors do a good job of explaining their work. There are, however, several typos which need to be fixed. Here is an incomplete list to give an idea of the typos that I found:
- “indirect-leveraging” in the abstract should be an em-dash instead of a hyphen
- Figure 3 says “trainformer”
- The caption in Table 1 (and in the supplement) say “parenthathese”
- Figure 5 caption says “validatiom”

There are also numerous grammatical issues, which I hope can be fixed, as well.

In terms of novelty, the paper is more limited. The novelties of the work (compared to previous works such as CVGAE and DMCG) are effectively to introduce a new format of input features, and use a variant of cross attention. Although these contributions are not trivial, the resulting model also does not demonstrate a significant improvement in performance over previous models (although they are somewhat more efficient).

**Strength And Weaknesses:**

The paper presents the authors’ main ideas in a largely clear manner, and it introduces and mostly justifies the choices made. The implementation details are fairly clear. The authors also present an interesting analysis on the loss curves which show a tradeoff between the reconstruction and variational losses, which is very appreciated. The authors are relatively thorough in their benchmarks with other works.

There were, however, some areas where the paper makes some design decisions which might not be optimal:
### Convolving over the input tensor with both atom and bond features can be problematic
The input tensor that is constructed is very heterogeneous, because it contains both atom feature vectors and bond feature vectors (side note: presumably, the atom feature vector is padded with 0s in order to make the two vectors equal size). Because the tensor is structured such that the atom features are along the diagonal, this means that a convolution operation over the tensor is not uniform. That is, consider an $N \times 3 \times 50$ convolutional kernel. The entry at $(0, 1, 0)$ will be multiplied against an atom feature in the first sliding window, and then it will be multiplied against a bond feature (which is twice the magnitude of the atom feature) in the next sliding window. This leads to the same entries in the kernel to experience entries in the tensor that are of varying magnitudes and meanings, which is suboptimal.
### Convolution is not particularly natural for this kind of input
Firstly, the input tensor is symmetric about the diagonal so the convolution sees redundant information. Combined with the issue brought up above, the redundancy structure is different for every sliding window (i.e. sometimes entries near the top and bottom are redundant, and other times entries near the middle are redundant).

Secondly, because the molecules are variable sizes, the input tensor will need to be zero padded to a maximal size. This means that the same kernel will also experience zeros when there is padding. This can also lead to confusion because zero is also a meaningful value in the input tensor (which is mostly one-hot encoded)
### Kabsch alignment shouldn’t be necessary
The use of Kabsch alignment (or alignment in general) should not be necessary. Transformers retain the ordering of the input vectors/tokens. That is, the $i$th input token is mapped to the $i$th output token through the attention mechanism. This means that the use of alignment should not be needed, as each atom/token can be mapped to an output token.
### Variation of cross attention is used without much justification
This is more of a minor point, but cross attention typically combines the key and value of one sequence with the query of another sequence. This work computes the key and query of the 2D tokens with the value of the 3D tokens. This is somewhat distinct from the traditional cross-attention mechanism. It would be good to have a justification for why this might work better than traditional cross attention.

**Summary Of The Paper:**

This paper illustrates a method for how to train a neural network to predict 3D conformers of small molecules starting from their 2D structures. Building on previous works such as CVGAE, the authors propose a few tweaks which are meant to improve the models’ performance: 1) use a single tensor to represent both atom and bond information, and featurize the single tensor into atom-level vectors using a 1D convolution operation on the tensor; and 2) use a transformer architecture and combine 2D and 3D representations during training using a variant of cross attention. The authors demonstrate that with their method, they can achieve performance comparable to other similar SOTA methods.

**Summary Of The Review:**

The paper (other than the grammatical issues and typos) is well written, and flows nicely. It is informative and the figures are generally very enlightening. The benchmarks are decently well done, as well. However, the paper demonstrates limited novelty in the technical space, as the proposed tweaks to existing work are either questionable or small. The result of these tweaks also do not seem to lend to significantly improved performance beyond the existing methods.

---

> ### Author Response · Authors · 2022-11-16
> **Response to reviewer 4X5j--Part 2**
>
> ### the input tensor is symmetric about the diagonal so the convolution sees redundant information
>
> Thanks for this comment. Please consider this problem from the perspective of information aggregation in a $N\times 3$ window. Taking the first $N\times 3$ window as an example, a $N\times 3$ kernel “sees” the complete connection pattern of the second atom (the focal atom) on the diagonal; that is how the second atom connects to its two immediate neighbours and all other atoms in a molecule. The convolution operation then aggregates information from all other atoms onto the second atom according to this specific connection pattern. The generated feature token contains information specific to the second atom. Let us slide the kernel to the last $N\times 3$ window, this time, a $N\times 3$ kernel “sees” a different connection pattern of the second last atom (the focal atom) on the diagonal. Likewise, the generated feature token contains information specific to the second last atom.
>
> Due to symmetry, one may notice that the feature values in the bottom left corner of the first window are the same as those in the top right corner of the last window. We figure this is the reason why you perceive them as redundant. However, these values, though identical, have a completely different meaning to a focal atom in a different sliding window. An intuitive explanation is **in a molecule, the influence of a substructure exerts on a hydrogen atom is completely different to the influence it exerts on another carbon atom, despite the feature values representing this substructure are the same for both atoms**. It is also worth noting that, the kernel weights that convolve with the same features values in the bottom left corner and top right corner are different. This is actually desirable as these feature values need to be weighted differently for different focal atom with different connection pattern. In summary, the information aggregated in each sliding window is unique and specific to a different focal atom; therefore it is not redundant.
>
> ### because the molecules are variable sizes, the input tensor will need to be zero padded to a maximal size. This means that the same kernel will also experience zeros when there is padding. This can also lead to confusion because zero is also a meaningful value in the input tensor (which is mostly one-hot encoded).
>
> Thanks for this excellent question. Padding input sequence of varying length is a standard practice in language modeling; for instance, when a transformer is applied to language translation, the input sentences can be of any length and so are the output sentences. Feature tokens generated from the padded areas will be masked out in both self-attention (using masked self-attention) and final loss computation. Therefore, these tokens do not contribute to parameters update. **For our application, we can easily determine the maximum number of atoms in a molecule for every training sample and identify those feature tokens generated from the padded region. Then, we mask out their values for all self-attention layers (also using masked self-attention), and for computing the Kl loss and RMSD loss. They do not contribute to kernel parameter update, and therefore there is no confusion**.
>
> ### Kabsch alignment shouldn’t be necessary
>
> We agree with the reviewer that the **alignment of the ordering** of atoms is not needed. However, the purpose of the [Kabsch](https://en.wikipedia.org/wiki/Kabsch_algorithm) alignment is **not aligning the ordering of atoms**, but rather **is aligning conformations in 3D space**. Imagine two identical conformations of a molecule in 3D space with different orientations (created by a random rotation and translation of one with respect to the other), the RMSD loss between these two identical conformations is not zero because their coordinates are not aligned. If we rotate one conformation again, the RMSD error also changes. This is not a desired behavior, as the RMSD loss between two conformations should be independent of their orientations. The Kabsch alignment achieves exactly this. It determines an optimal rotation matrix to align the coordinates of a predicted conformation as close as possible to those of the ground truth conformation. The resulting RMSD loss computed is therefore invariant to rotation and translation (or SE(3) invariant). Due to its importance, **Kabsch alignment is universally adopted in all baselines** compared in Table 1 of our submission.
>
> ### Variation of cross attention is used without much justification
>
> Thanks for this comment. We would kindly ask reviewer to please look at a paragraph titled *Intuition behind the modified attention* in section 2.4 of the original and revised submission, where we fully justify why the modified self-attention (the variation of cross attention referred by the reviewer) is necessary for achieving a good performance.

---

> > ### Comment · Reviewer_4X5j · 2022-12-09
> > **Reply**
> >
> > Thank you for your clarifications!
> >
> > ### Convolving over the input tensor with both atom and bond features
> >
> > My apologies for using misleading terminology. The issue I was pointing out was specifically the heterogeneity observed as the filter is slid along the input tensor. For entries _on_ the diagonal, the first 12 features describe the atom. For entries _off_ the diagonal, the first 12 features have twice the magnitude and they mean something different (they are summed atom features). Similarly, for entries _on_ the diagonal, the last 11 features are 0. For entries _off_ the diagonal, the last 11 features are meaningful bond features. So as the kernel is slid across the input tensor, any single weight/parameter will see values of different meanings depending on where the kernel currently is. Although it is good to know that there isn't an obvious observed issue, this sort of heterogeneity makes the use of convolutions rather questionable.
> >
> > ### Redundancy
> >
> > I can see an argument for why redundancy can be helpful, and how the symmetry can help the model learn the importance of certain atoms (although I am not entirely convinced of it yet). However, the symmetry structure changes as the kernel is slid across the input tensor. This is very much related to how the input tensor isn't really natural for convolutions, because the kernel is static, yet the structure in the input changes so much at different positions.
> >
> > ### Zero padding
> >
> > I see, thank you for clarifying the masking scheme. I did not come across this in the manuscript, so I would suggest making it more obvious.
> >
> > ### Kabsch alignment
> >
> > Thank for for explaining!
> >
> > ### Cross attention
> >
> > Again, this is a very minor point (and not one that I think is critical at all). The main question I had was why the cross attention mechanism is _different_ from some other works. Most cross attention applications combine the key and value of one sequence with the query of another sequence. Here, you combine the key and query of the 2D tokens with the value of the 3D tokens. So it's a little different. I would have liked an explanation as to why this scheme is better than the "normal" cross attention scheme.

---

> > > ### Author Response · Authors · 2022-12-09
> > > **Our design choices are supported by extensive experimental evidence**
> > >
> > > Thanks for your reply. Our design choices including the tensor representation, 1D convolution and modified self-attention are centred around **a main goal--a simple and efficient generative model for molecular conformation generation**. In terms of simplicity, the design choice of encoding a molecular graph with a tensor representation followed by a 1D convolution has enabled a unmodified transformer architecture with a significantly less number of parameters and without pretraining to achieve SOTA performance as compared to the recent 2 SOTA methods. In terms of efficiency, the sensible integration of two encoder inputs via the modified self-attention in a VAE framework allows a direct conformation generation from a 2D molecular graph in a single step.
> > >
> > > **The design choices made above successfully achieve the main goal. The resulting promising performance is also supported by extensive experimental evidence. In light of the strong experimental evidence, we are reluctant to agree with your qualitative assessment of "rather questionable" design choices**.
> > >
> > > Again, we acknowledge your concern that heterogeneity of convolution might be a potential issue. However, we are not sure if it is an actual issue as there is no experimental evidence supporting it. We will explore this problem further in our future work.

---

> ### Author Response · Authors · 2022-11-16
> **Response to reviewer 4X5j--Part 1**
>
> We thank reviewer 4X5j for providing constructive feedback. We deeply appreciate your comment “*The paper (other than the grammatical issues and typos) is well written, and flows nicely. It is informative and the figures are generally very enlightening*”. While we hope our general responses address your concern over limited novelty, we provide further responses to address the following 5 specific comments. Additionally, we have also fixed all the mentioned typos and thoroughly proof-read the manuscript to correct all grammatical errors.
>
> ### Convolving over the input tensor with both atom and bond features can be problematic
>
> Thank you for your keen observation. The convolution operation is in fact uniform, as the feature type in each channel of the tensor is the same regardless whether the location of a cell is on-diagonal or off-diagonal. **This is because like the ordering of the RGB channels of a colored image stay the same at every pixel location, the ordering of the channels of the input tensor is always the same in each of the $N\times N$ cells of the tensor**.
>
> To make this point more clear, we invite reviewer 4X5j to pay attention to figure 1 of our submission, where we have shown examples of 2 feature vectors on diagonal and off diagonal respectively. **Although we denote the feature vectors on-diagonal the *atom feature vectors* and those off-diagonal the *bond feature vectors*, the length and structure of these two types of vectors are exactly the same**. For both types in the GDR tensor, there are 5 blocks (3 blocks for G tensor) of channels **stacked in the exact same order as following**;
>
> • atom feature channels (atom type, charge, and chirality);
>
> • atom coordinate channels (coordinate channels are excluded in the G tensor);
>
> • bond type feature channels (bond type and normalized bond length);
>
> • Euclidean distance channel (pairwise distance channel is excluded in G tensor);
>
> • other bond type feature channels (bond stereo-chem type and bond ring size).
>
> There are no bond and distance features on-diagonal. The corresponding channels are filled with zeros. Similarly, as there are no coordinate feature in the off-diagonal section, these channels are filled with zeros as well. As the channel ordering/structure is uniform/same across all cells, **it is impossible for the same entries (0,1,0) of a kernel to be multiplied by a bond type feature in one stride and then by an atom type feature in the next stride**. In other words, if the entries (0, 1, 0) attend to the atom type channels, it will always do so for every sliding window. Therefore, each entry of the kernel will always be multiplied by a channel with the same meaning.
>
> Reviewer 4XHj is correct about the varying magnitude between atom features on-diagonal and off-diagonal due to summation of atom features (we do not sum bond features, as there is only one set of bond features per atom pair). We agree that this might be a potential issue; however, we did not observe any instability caused by it during training for all our experiments.

---

### Official Review · Reviewer_iXjq · 2022-10-24

**Confidence:** 2
**Correctness:** 1
**Technical Novelty And Significance:** 3
**Empirical Novelty And Significance:** 3
**Recommendation:** 8

**Clarity, Quality, Novelty And Reproducibility:**

The paper is well-written and the proposed method seems novel and interesting to the community. For the model to be fully reproducible, I detailed the experimental setup and the code should be published in the camera-ready version.

**Strength And Weaknesses:**

Strength:
the idea is to combine Atom and edge features into a single input by adding an additional dimension to
the adjacency matrix, making it a tensor where the diagonal section of the tensor holds the atom features sounds interesting.

Weakness: the models seem to be really big (because of the input tensor)

**Summary Of The Paper:**

The paper tackles the generation of 3D conformations of a molecule from its 2D graph. They propose to encode a molecular graph using
a fully connected and symmetric tensor. They use the standard VAE framework, where they build two input tensors with one encoding only the 2D molecular graph and the other also encoding 3D coordinates and distance. Both tensors go through the same feature engineering step and the generated feature vectors are fed through two separate transformer encoders. The output of these two encoders is then combined in an intuitive way to form the input for another transformer encoder for generating confirmation directly.

**Summary Of The Review:**



Some detailed comments:

1. I wonder why the model needs two conditional encoders, one conditioned on the graph only and another one conditioned on both the graph and coordinate, why not directly use the latter since it has also graph information?

2. I am a bit confused about section 2.3 where the second main idea is explained.

---

> ### Author Response · Authors · 2022-11-16
> **Response to reviewer iXjq**
>
> We appreciate reviewer iXjq' encouraging words. In addition to our general response, we provide detailed responses to the following 3 comments.
>
> ### The models seem to be really big (because of the input tensor)
>
> Thank you very much for pointing this out. As explained in general response 2, TensorVAE has a significantly less number of parameters as compared to the current SOTA models.
>
> ### I wonder why the model needs two conditional encoders, one conditioned on the graph only and another one conditioned on both the graph and coordinate, why not directly use the latter since it has also graph information?
>
> Thanks for the excellent question. We follow the same variational autoencoder framework as in ConfVAE and CGVAE to train our network. A variational autoencoder in this context assumes generating a random conformation can be achieved using a decoder that depends on two input features, a feature encoding the 2D graph $G$ and a latent feature $z$. The latent feature $z$ is the probabilistic component contributing to the variation in conformation. While the G encoder is used to encode the 2D graph, the GDR encoder approximates the posterior distribution of the latent feature $q_w(z|R,G)$. During training, one of the objectives is to minimize the KL-divergence between $q_w(z|R,G)$ and the prior $p(z)$, where $p(z)$ is often assumed to be a spherical Gaussian distribution. Minimizing this objective can be considered as a regularization technique that facilitates model generalization. After training converges, we assume $q_w(z|R,G)$ is close enough to $p(z)$. Therefore, at inference time, we sample $z$ directly from $p(z)$ and the GDR encoder is no longer needed. In summary, we only need both encoders during training as the objective associated with GDR encoder acts as a regularization. At inference time, we only need the G encoder and a random z sampled from $p(z)$, so we can generate conformation directly from a 2D molecular graph.
>
> ### I am a bit confused about section 2.3 where the second main idea is explained
>
> The essential information of section 2.3 is summarized in figure 2. We intend to illustrate that by extending a $3\times3$ kernel to a $N\times3$ one, we are able to substantially expand its field of view. As a result, it can always see the complete connection pattern of a focal atom. We also show that, due to this expansion, only 1D convolution is allowed; that is the kernel can only slide from left to right. There are multiple (256) kernels being applied at each stride. That is why we can generate a feature token (dimension 256) for a focal atom per stride.

---

### Official Review · Reviewer_n6uh · 2022-10-27

**Confidence:** 4
**Correctness:** 3
**Technical Novelty And Significance:** 2
**Empirical Novelty And Significance:** 2
**Recommendation:** 5

**Clarity, Quality, Novelty And Reproducibility:**

The presentation is clearly written, and the work appears to be of sufficient quality.

The feature engineering is marginally original (see weaknesses above). Combining 3D latents as values with 2D embeddings as keys and queries in the VAE formulation appears to be a novel way of learning to generate 3D confirmations from 2D graphs.

The authors claim everything is straightforward and provide no code.

**Strength And Weaknesses:**

Strengths:
- Simple architecture and featurization
- Original use of two encoders for 2D and 3D features in the VAE formulation that allows for 3D conformation generation from 2D graphs
- (Near) state-of-the-art results for conformation generation (and QM9 property prediction)

Weaknesses:
- It is not clear that the proposed method of producing atom-tokens via 1D convolution on the input tensor is necessary. Any number of possible aggregation steps could have been used, many of which would likely have yielded similar results. One could also imagine performing a Tucker decomposition and using singular values as tokens, etc.
- Moreover, it's not even clear if the tensor formulation is needed. The 1D convolution aggregates information about the radius-1 atomic environment (including virtual bonds). One could use any radius-1 atomic-environment hash as tokens with the proposed featurization, which would likely yield comparable results.

**Summary Of The Paper:**

The authors propose TensorVAE, a relatively simple model for generating 3D conformations from 2D molecular graphs. TensorVAE employs 1) a unique feature engineering step that represents each molecule as a tensor (with or without 3D coordinates and distances), and 2) a VAE with two transformer encoders, one that encodes the graph (using the tensor input without 3D information) and the approximate posterior that produces latents from the 3D information, and a transformer decoder for the likelihood, where keys and queries come from the 2D graph representation, and values are the latents from the 3D posterior encoder. The loss is a standard roto-translation invariant loss.  The authors show that, using standard transformer architectures and training procedures, TensorVAE performs comparably to the best current methods at conformation generation using the GEOM dataset, and argue that TensorVAE is much simpler than the comparable models due to superior feature engineering.

**Summary Of The Review:**

The paper extends existing methods and molecular featurizations to achieve near state-of-the-art results for conformation generation. The paper spends a lot of time arguing for the superiority of the feature engineering used, but it is not at all clear that the results depend on that featurization. The authors could greatly improve this paper by demonstrating the value of the feature engineering beyond performance on benchmarks through ablation studies (e.g. removing aspects of the features) and studies of other featurizations that are not conflated with the tensor design (e.g., radius-1 atom environments).

---

> ### Author Response · Authors · 2022-11-16
> **Response to reviewer n6uh**
>
> We thank reviewer n6uh again for your feedback. We provide further response to the following 4 comments.
>
> ### It is not clear that the proposed method of producing atom-tokens via 1D convolution on the input tensor is necessary
> Please refer to our general response 3 and ablation studies for the answers to this question.
>
> ### Moreover, it's not even clear if the tensor formulation is needed. The 1D convolution aggregates information about the radius-1 atomic environment (including virtual bonds).
> As we have explained in our general response 3, the information aggregated by the 1D convolution operation extends far beyond radius-1 atomic environment. We have also demonstrated in ablation studies that this type of aggregation does lead to improved performance.
>
> ### The paper spends a lot of time arguing for the superiority of the feature engineering used, but it is not at all clear that the results depend on that featurization. The authors could greatly improve this paper by demonstrating the value of the feature engineering beyond performance on benchmarks through ablation studies (e.g. removing aspects of the features)
>
> We deeply appreciate this suggestion. We have concretely demonstrated in our general responses and additional ablation studies that the proposed feature engineering is effective. It enables a vanilla transformer architecture with a significantly less number of parameters, without specialized loss function design, and without pre-training, to match or exceed the performance of Uni-Mol and DMCG.
>
> ### Compared to models with featurization other than a tensor
>
> In fact, Uni-Mol serves as a concrete example of applying a transformer backbone on a different featurization. As has been explained in the General response 2, Uni-Mol treats a molecule as a collection of point clouds rather than converting it to a tensor. It also introduces a pair interaction matrix to facilitate information aggregation between atom tokens. Notice again since the self-attention can be considered as a fully-connected GNN, a 1-hop (or a “radius-1 atomic environment”) message passing also achieves global information aggregation. However, with this featurization, Uni-Mol needs pre-training on 209M conformations and a significantly larger and modified transformer to match our performance. Similarly, DMCG's backbone is also an attention-based architecture. Its inputs are separated atom and bond features. It uses a GNN to achieve 1-hop information aggregation. With this featurization, DMCG's performance is inferior to the proposed TensorVAE even with $10\times$ more model parameters.

---

### Author Response · Authors · 2022-11-16
**To all reviewers**

We would like to express our most sincere gratitude to all reviewers for your insightful and constructive comments. We have carefully studied your comments and have striven to address all of them in the revised submission. We first summarize the major improvements made to address your concerns.

• **Results update**: the Uni-Mol baseline results we cited are from an outdated [Version 1](https://chemrxiv.org/engage/chemrxiv/article-details/628e5b4d5d948517f5ce6d72), from which we concluded the performance of TensorVAE is only near SOTA. In its latest release [Version 3](https://chemrxiv.org/engage/chemrxiv/article-details/6318b529bada388485bc8361), the revised Uni-Mol baseline results are on par with ours, with both achieving SOTA.

• **Comparison of model parameters**: we have compared the number of model parameters among TensorVAE and the current SOTA models (Uni-Mol and DMCG). TensorVAE has a significantly less number of parameters.

• **Ablation studies**: we have added 3 additional ablation studies to further demonstrate the effectiveness of the proposed input feature engineering method.

We detail these improvements in 3 general responses. Additionally, we also provide responses to your specific questions that are not addressed in the general responses. We have highlighted the changes made in the revised submission.

**We will release the source code of TensorVAE via GitHub immediately after the conclusion of the double blind review**

---

> ### Author Response · Authors · 2022-11-16
> **Ablation studies**
>
> ### Why is 1D convolution necessary
>
> The primary reason for why 1D convolution is needed is that each $N\times 3$ kernel's “field of view” encompasses the complete connection pattern (including both chemical and virtual bonds) of a focal atom and its two nearest neighbors, whereas a conventional $d\times d$ ($d<N$) only sees a partial pattern. In the original submission, we have already demonstrated the inferior performance of applying a $3\times 3$ kernel on the proposed tensor through experiments with a model referred to as the “NaiveUNet” which itself is considered as an ablation study. Here, we provide a more detailed analysis regarding why NaiveUNet does not provide satisfactory performance.
>
> We observe that when applying a $3\times 3$ kernel filter to the top left region of the proposed tensor, its “field of view” only includes a focal atom, its two neighboring atoms and how the focal atom is connected to them. Firstly, this corresponds exactly to what reviewer n6uh has described as the “radius-1 atomic-environment”. It only achieves a 1-hop information aggregation. Secondly when the $3\times 3$ kernel moves to an off-diagonal part of the tensor, where most connections are virtual bonds (as atoms of a molecule are often sparsely connected), information aggregation occurs mostly between atoms that are not chemically connected and is therefore less meaningful. For these two reasons, NaiveUNet performance is the worst as shown in table below. It also performs much worse than similar distance-based approaches such as CGCF and ConfVAE (as shown in Table 1 of the revised submission) which also rely on 1-hop information aggregation enabled by GNNs. Therefore, extending the kernel size to $N\times 3$ is crucial to a good conformation generation performance.
>
> ### What happens if we remove all virtual bonds
>
> Notice that if we remove all the virtual bonds in each column and still run a $N\times3$ kernel through the tensor, its “field of view” is a “radius-2 (2-hop) atomic-environment” (because the focal atom can “see” how neighboring atoms are chemically connected to all their direct neighbors). Another observation is that after removing all virtual bonds, each column does not correspond to a fully-connected GNN. Therefore it no longer enables a global information aggregation. The conformation generation results of this variant of TensorVAE on Drugs dataset is shown as **as TensorVAE abla1** in table below. It is observed that due to local-only information aggregation as a result of removing all virtual bonds (and related atom features), the performance is worse than the complete TensorVAE version.
>
> ### What happens if a $N\times1$ kernel is used
>
> The final ablation study concerns with using a $N\times1$ kernel with a smaller field of view as compared to that of a $N\times3$ kernel. Its performance on Drugs dataset is shown as **TensorVAE abla2** below. It performs slightly better than the ablation removing all virtual bonds. The reason is that though its field of view is smaller, it still achieves a global information aggregation for the focal atom.
>
> | Model | COV mean | COV median|MAT mean|MAT median|
> | -----------| ----------- | ----------- | ----------- | ----------- |
> | NaiveUNet | $52.14\pm1.48$|$51.69\pm1.17$|$1.4322\pm0.0247$|$1.3861\pm0.0173$|
> | TensoVAE abla1 | $90.72\pm1.54$|$99.51\pm0.64$|$0.8748\pm0.0161$|$0.8619\pm0.0214$|
> | TensoVAE abla2 | $91.04\pm1.21$|$99.74\pm0.42$|$0.8706\pm0.0131$|$0.8561\pm0.0204$|
> | TensorVAE| $93.34\pm1.17$|$99.9\pm0.31$|$0.8074\pm0.0135$|$0.7927\pm0.0186$|

---

> ### Author Response · Authors · 2022-11-16
> **General Response 3: global information aggregation**
>
> A geometric interpretation of GNN's message passing layer is it aggregates information between atoms (and their bond) that are 1-hop (radius-1) away. With $L$ layers, information from atoms that are $L$-hop (radius-L) apart can be aggregated. Here, we define a global information aggregation as the $N^{th}$-hop (radius-N) aggregation with $N$ being the total number of atoms, where each atom is able to aggregate information from its farthest neighbour.
>
> It is worth noting that for a fully-connected GNN, a 1-hop message passing can already achieve this global information aggregation. Transformer's self-attention can be considered as a type of fully-connected GNN. However, a vanilla transformer can only aggregate features from each token/atom; if edge features are not included, they need to be incorporated somehow through additional inputs (e.g. the pair interaction matrix of Uni-Mol). The primary reason motivating the creation of the fully-connected tensor representation is we want each generated token to contain both atom and bond features, such that we can eliminate the pair interaction or bond matrix. To achieve this, we fill each column of the fully-connected tensor with;
>
> 1. focal atom features
>
> 2. chemical and virtual bond features indicating how the focal atom is connected to all other atoms
>
> 3. atom features of all other atoms, since for each off-diagonal cell, we also include atom features of the connected (via virtual or chemical bond) atom pair
>
> As nicely pointed out by reviewer 8zkT, **running a $N\times1$** (the actual kernel size is $N\times1\times C$, where C is the channel depth. We omit C for brevity of presentation) kernel filter on the proposed tensor **is conceptually similar to achieving a global information aggregation with a fully-connected GNN**. By increasing kernel width to 3, the aggregation window also includes **global information from two immediate neighbours**. In other words, a single operation of a $N \times 3$ kernel aggregates global information from three nearby atoms which extends far beyond a “radius-1 atomic environment”.
>
> More interestingly, when multiple kernels are applied simultaneously to the same $N\times 3 \times C$ region, each kernel is free to choose whichever group of atom/bond features to attend to depending on its kernel weights. **This resembles the multi-head attention mechanism of a transformer, where each kernel(head) contributes to a portion of the generated feature token**. We believe the effective global information aggregation driven by these two (tenor representation + 1D Conv) simple yet intuitive ideas is the main reason why the proposed TensorVAE achieves SOTA with much less number of parameters.

---

> ### Author Response · Authors · 2022-11-16
> **General Response 2: Further analysis and evidence to support novelty and contribution**
>
> We further demonstrate the novel and significant contribution of the proposed TensorVAE through a more detailed comparison to both DMCG and Uni-Mol.
>
> We first compare the feature engineering method. DMCG adopts a feature engineering method similar to other compared GNN-based baselines (ConfGF, GeoDiff, etc.). On the other hand, Uni-Mol treats a molecule as a collection of nodes (or a point cloud). To capture the interaction between atoms, Uni-Mol has also introduced a pair interaction matrix which is first computed as an affine transformation of the distance matrix, and is then updated and maintained throughout the self-attention layers. **For both methods, information aggregation among atoms and bonds is not a part of their input feature engineering.**
>
> In terms of backbone architecture, DMCG utilizes a highly modified GNN block for information aggregation. Noticeably, a bond representation matrix is also maintained throughout all layers. Inspired by EvoFormer's “Structure Module”, the GNN block also outputs intermediate conformations that are iteratively refined through every decoder layer. Furthermore, DMCG combines a gated attention network (GATv2) with the GNN block for more effective information aggregation. For Uni-Mol, its backbone is a specialized transformer encoder able to take pair interaction matrix as an additional input. Atom features are treated as tokens in the first layer and are used to compute Key, Query and Values matrices. A modified self-attention mechanism also inspired by Alphafold's EvoFormer is used to aggregate pair information onto the atom features and vice versa. **A feature common to both DMCG and Uni-Mol is that as information aggregation is not a part of their input feature engineering, they rely exclusively on complex neural network architecture to achieve effective information aggregation.**
>
> In terms of output head and loss function, for Uni-Mol, an additional SE(3) equivariant output head is needed to generate output coordinates. For DMCG, a symmetric permutation invariant loss function in addition to the Kabsch alignment function is required to achieve a good performance. All these extra components further add complexity to the model architecture.
>
> In clear contrast, the proposed input feature engineering achieves a global information aggregation among atoms and bonds at the input level (more explanation of this in General response 3). Unlike Uni-Mol's input tokens, an atom token generated by this feature engineering contains both atom and bond information. **A significant benefit of such design is that it is no longer necessary to maintain a pair interaction matrix through every layer. This enables a drastic reduction of the number of model parameters.** This fact is consolidated by comparing the number of model parameters among DMCG, Uni-Mol and TensorVAE (TensorVAE has less number of parameters during inference time as the GDR encoder is not needed), as shown in table below. The number of model parameters for DMCG is obtained from [here](https://github.com/DirectMolecularConfGen/DMCG). For Uni-Mol, it is obtained from [here](https://github.com/dptech-corp/Uni-Mol)
>
> | Model | # of parameters |
> | --- | ----------- |
> | DMCG | 128M |
> | Uni-Mol | 47.81M |
> | TensorVAE training | 11.5M |
> | TensorVAE inference | 6.65M |
>
> **More concretely, we summarize 4 major differences between TensorVAE and the other two SOTA models**;
>
> 1. A vanilla transformer architecture is used without task-specific modification;
>
> 2. A simple Kabsch alignment loss function is used;
>
> 3. A generative model with at least $4\times$ less number of parameters as compared to Uni-Mol and $10\times$ less as compared to DMCG;
>
> 4. There is no pre-training for TensorVAE, whereas Uni-Mol is first pre-trained on 209M conformations collected from multiple public datasets and then fine-tuned with 200,000 GEOM conformations.
>
> Each of these differences puts TensorVAE at a significant disadvantage in terms of model capacity, let alone combine them together. Yet, the proposed TensorVAE either matches or outperforms Uni-Mol and DMCG on a more challenging test-set that has $10\times$ more molecules (and conformations). **The only logical deduction for explaining this promising performance is the superiority of the proposed input feature engineering**. We further argue that achieving this feat is anything but a marginal contribution with limited novelty.
>
> Now that we have established the novelty of the input feature engineering, we will further demonstrate the necessity of the tensor representation and its accompanying 1D convolution through more detailed analyses in General Response 3, and then followed by 3 ablation studies.

---

> ### Author Response · Authors · 2022-11-16
> **General response 1: results update and attaining SOTA**
>
> We sincerely thank reviewer 8zkT for pointing out the differences between the results in Table 3 of Uni-Mol and in Table 1 of our original submission. **These differences are not typos, but are conflicting results from two versions of Uni-Mol, [Uni-MolV1](https://chemrxiv.org/engage/chemrxiv/article-details/628e5b4d5d948517f5ce6d72) and [Uni-MolV3](https://chemrxiv.org/engage/chemrxiv/article-details/6318b529bada388485bc8361)**. **The Uni-Mol baseline results in V3 were released on Sep 08, 2022 and are worse than those in V1. We were unaware of these changes at the time of submission, and have cited V1 baseline results for comparison**. We have updated Table 1 results in the revised submission with the latest results from Table 3 of Uni-MolV3.
>
> It appears that Uni-MolV1 baseline did not follow the molecule filtering condition in the ConfGF paper as outlined [here](https://github.com/DeepGraphLearning/ConfGF). With this condition, only molecules with a number of annotated conformations in a certain range are considered for testing (50 to 500 for QM9 and 50 to 100 for Drugs). Uni-Mol authors also identified DMCG and GeoDiff did not follow this condition either, and have rerun these baselines. We appreciate this effort by the Uni-Mol authors.
>
> We have strictly followed the test data generation procedure of ConfGF. In fact, all cited baselines have claimed to follow this procedure; that is selecting 200,000 conformations (top 5 conformations from 40,000 randomly selected molecules) for training, and then testing on 200 randomly selected molecules satisfying the aforementioned filtering condition. In the ConfGF paper, the total numbers of annotated conformations for 200 testing molecules are 22408 and 14324 for QM9 and Drugs respectively. In our sample of 200 molecules, there are slightly more testing conformations (23079 and 14396 respectively). The standard deviations of our original results (**denoted as TensorVAE1 in Table 1 of the revised submission**) were obtained by running 10 experiments with different seed for conformation generation on the same test-set.
>
> We concur with reviewer 8zkT's comment pointing out a single sample of 200 testing molecules is small. To address this, we sample a different set of 200 molecules for each of the 10 experiments. In total, **2000** testing molecules are selected for both QM9 and Drugs experiments, with **280,000** and **144,000** annotated conformations used for testing respectively. **In this setup, the number of testing conformations is more than 70% (140% for QM9) of that of training conformations**. The updated results are **denoted as TensorVAE2 in Table 1 of the revised submission**. It is observed that **the performance of the proposed TensorVAE is on par with Uni-MolV3. It also consistently outperforms DMCG across both datasets**. Thus, **our model achieves the SOTA performance**. Achieving a good performance on this much larger test-set verifies the generalization capability of the proposed TensorVAE. On a different note, **none of the cited baselines** have computed standard deviation of their results, despite the fact they all run a single test on 200 random molecules.
>
> In addition, we have also carefully checked the updated molecular prediction result of Uni-MolV3, and noticed the scaffold splitting method is used for obtaining train, valid, and test QM9 molecules. However, we followed the recommended random splitting found in Table 1 of the [MoleculeNet](https://pubs.rsc.org/en/content/articlelanding/2018/sc/c7sc02664a) paper. Scaffold splitting is more challenging as the test molecules contain completely different scaffold structure as compared to those in the training set.
>
> Uni-MolV3 also added results of two additional variants including one without pre-training and the other excluding the pair interaction matrix. The latter ablation reduces the backbone to a vanilla transformer similar to ours. In light of these new details, we rerun the experiments using the scaffold splitting and reported the updated results in Table 4 of the revised submission. Our results are admittedly worse than the complete Uni-Mol version with pre-training on 209M molecules. However, it outperforms the other two variants and 6 other methods with a significant margin. This new result still verifies the effectiveness of the proposed input feature engineering, as TensorVAE uses a vanilla transformer architecture with no pre-training.
>
> We acknowledge achieving SOTA only partially justifies the novelty of our work. Therefore, we provide further evidence supporting our contribution in General Response 2. In particular, **we show that such performance is achieved with at least $4\times$($7\times$ during inference) less number of parameters as compared to Uni-Mol, and $10\times$ less as compared to DMCG**. We also explain in depth why the proposed feature engineering is able to achieve this.

---

### Author Response · Authors · 2022-12-12
**Concluding remark for discussion phases**

We sincerely thank all reviewers for reviewing and providing constructive feedback to our work. Your comments have provided valuable insights that improve the clarity and help substantiate the novelty of the proposed TensorVAE model. To facilitate AC's decision making and for ease of potential readership, we summarize the major concerns raised in the two discussion phases and how we have addressed them, as following;

1. First and foremost, there is a common concern among reviewers regarding the novelty and contribution of the proposed tensor-based feature engineering method. To address this concern, we have emphasized that **the main contribution of our work is a simple and efficient generative model for molecular conformation generation.** Our simple model achieves the state of the art performance with at least **7 times less number of parameters**, with a **simple loss function** and **without any pretraining** as compared to the current 2 SOTA models. We argue that achieving this feat with a simple and intuitive method is quite a significant contribution.

2. The core module of the proposed TensorVAE is a CNN-based feature engineering method. There are doubts regarding the design choices we have made for encoding a molecular graph with a fully-connected tensor representation. To address this concern, we conducted 4 additional ablation studies (see [here](https://openreview.net/forum?id=TtMJJWG_J1j&noteId=6QfflKGoGhC) and [here](https://openreview.net/forum?id=TtMJJWG_J1j&noteId=Z2zrvnSBN8)) that concretely demonstrate that the choice of feature representation and engineering with a 1D convolution operation using  $N \times 3$ kernels does lead to significant performance improvement. The legitimacy of this design choice is fully supported by extensive experimental evidence.

3. There are also questions relating to the similarity between a fully-connected GNN and the CNN-based feature engineering method. To address this, we have demonstrated the mathematical equivalence (see [here](https://openreview.net/forum?id=TtMJJWG_J1j&noteId=tqBEZVymWg)) between the information aggregation of a fully-connected GNN and that achieved by running a $1 \times 1$ convolution operation over the proposed tensor representation. **A fully-connected GNN corresponds to the simplest form of our feature engineering and has the least model capacity and expressive power for information aggregation**. Establishing this fact also leads us to conclude that the ability of the proposed TensorVAE to learn the difficult task of conformation generation improves with the expressive power of its information aggregation mechanism. The benefit of permutation invariance brought by a $1 \times 1$ kernel is not as significant as that of the increased model capacity and expressive power brought by using a larger $N \times 3$ kernel.

In conclusion, we believe that all major concerns are addressed and the novelty of our method is concretely supported by extensive experimental evidence.

---

### Decision · Program_Chairs · 2023-01-20

**Decision:**

Reject

**Justification For Why Not Higher Score:**

There were too many points of criticism that could not be resolved convincingly in the rebuttal.

**Justification For Why Not Lower Score:**

N/A

**Metareview: Summary, Strengths And Weaknesses:**

For this paper, we had three (slightly) negative reviews and one clearly positive review.  Summarizing all reviews, the rebuttal and the following discussion phase, I finally came to the conclusion to vote for rejection of this paper. Let me explain this in more detail:

On the positive side, all reviewers mentioned that the motivation is clear,  the general idea is interesting and the underlying problem is of high importance. On the other hand, several points of criticism have been raised, such as:
- the quality and interpretation  of (some of) the comparison experiments was criticized by some reviewers. Some of these concerns could be addressed in the rebuttal, so I think this is only a minor point of criticism. The more important points are:
-  novelty and significance of the paper are limited, a conceptual contribution is not clearly visible. I think, this is a very valid concern that could not be addressed in the rebuttal. Even though some (of the more negative) reviewers pointed out that they were positively impressed by the detailed rebuttal in general, they still wanted to keep their negative scores.
- the use of convolutions is somewhat questionable, due to heterogeneity issues in the input tensor. I agree with some of the reviewers that this is a critical concern that questions the general idea of using convolutions in such settings.  Also this issue could not be fully resolved in the rebuttal, and many open questions remained.

The only positive review included only somewhat general comments and the reviewer declared a relatively low confidence. Further, this reviewer finally did not clearly champion this paper for acceptance.